# DipLLM: Fine-Tuning LLM for Strategic Decision-making in Diplomacy

Kaixuan Xu [* 1 2]  Jiajun Chai [* 2 1]  Sicheng Li [2 1]  Yuqian Fu [2 1]  Yuanheng Zhu [2 1]  Dongbin Zhao [2 1]

## Abstract

Diplomacy is a complex multiplayer game that requires both cooperation and competition, posing significant challenges for AI systems. Traditional methods rely on equilibrium search to generate extensive game data for training, which demands substantial computational resources. Large Language Models (LLMs) offer a promising alternative, leveraging pre-trained knowledge to achieve strong performance with relatively small-scale fine-tuning. However, applying LLMs to Diplomacy remains challenging due to the exponential growth of possible action combinations and the intricate strategic interactions among players. To address this challenge, we propose DipLLM, a fine-tuned LLM-based agent that learns equilibrium policies for Diplomacy. DipLLM employs an autoregressive factorization framework to simplify the complex task of multi-unit action assignment into a sequence of unit-level decisions. By defining an equilibrium policy within this framework as the learning objective, we fine-tune the model using only $1.5\%$ of the data required by the state-of-the-art Cicero model, surpassing its performance. Our results demonstrate the potential of fine-tuned LLMs for tackling complex strategic decision-making in multiplayer games.

## 1. Introduction

Multiplayer games have long been a cornerstone of AI research, with classic examples like chess (Silver et al., 2017), Go (Silver et al., 2016; 2017), and poker(Moravčík et al., 2017; Brown & Sandholm, 2018) with at most thousands of actions per state. In contrast, Diplomacy presents a significantly more complex challenge, with a combinatorial

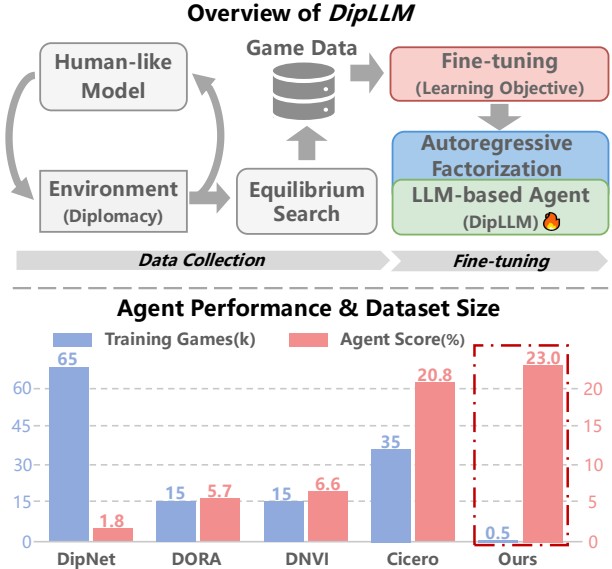

Figure 1. An overview of our agent, DipLLM and performance of various agents in a population.

action space that can exceed $10^{64}$ possible choices per turn. This complexity stems from the game's mechanics, where each player controls up to 34 units, each with around 26 possible actions. The exponential growth in possible action combinations, coupled with the intricate interactions among players, makes Diplomacy an challenging environment for advancing AI in strategic decision-making. Traditionally, success in Diplomacy has relied on equilibrium search methods to generate equilibrium policies (Zhu & Zhao, 2022; Jacob et al., 2022; Wongkamjan et al., 2024; Lu et al., 2025). While effective, as illustrated in Figure 1, these methods require large amounts of game data[1] to train models, limiting their scalability and applicability to other domains.

In parallel, Large Language Model (LLM)-based agents have gained significant attention in AI research due to their powerful general reasoning abilities. These agents have shown strong performance in a variety of domains, including personal assistants (Li et al., 2024), robotics (Chai et al., 2025; Chen et al., 2025b) and games (Jin et al., 2024; Guan

---

*Equal contribution [1]School of Artificial Intelligence, University of Chinese Academy of Sciences, Beijing, China [2]State Key Laboratory of Multimodal Artificial Intelligence Systems, Institute of Automation, Chinese Academy of Sciences, Beijing, China. Correspondence to: Yuanheng Zhu <yuanheng.zhu@ia.ac.cn>.

*Proceedings of the 42nd International Conference on Machine Learning*, Vancouver, Canada. PMLR 267, 2025. Copyright 2025 by the author(s).

[1]The data requirements for DORA, DNVI, and Cicero during training are estimated based on GPU usage, as their respective papers do not provide exact numbers.

et al., 2024). Typically, LLM-based agents rely on prompt engineering, where the input text is adjusted to enhance performance for specific tasks without modifying the model's weights. Although this approach can yield short-term performance gains, it remains constrained by the inherent limitations of the foundational model, often leading to suboptimal decision-making in complex scenarios (Xu et al., 2024). Recent advancements (Zhai et al., 2024; Bai et al., 2024) suggest that fine-tuning LLMs on small, domain-specific datasets can significantly improve their performance by optimizing policy distributions. While this method has proven effective in environments with smaller action spaces, it struggles in complex multiplayer games like Diplomacy due to the exponential growth of action combinations and intricate strategic interactions among players. This raises a critical question: **Can we fine-tune an LLM-based agent to learn equilibrium policies that outperform traditional methods in Diplomacy?**

To address this challenge, we propose DipLLM, a fine-tuned LLM-based autoregressive factorization agent that learns equilibrium policies for Diplomacy (Figure 1). Our agent leverages an LLM-based autoregressive factorization framework to decompose the complex task of assigning actions to all units into sequential sub-tasks, each focusing on a single unit's action. This decomposition significantly reduces the combinatorial complexity of the action space, offering a manageable foundation for fine-tuning. Within this autoregressive factorization framework, we define a learning objective that approximates the Nash equilibrium and provide theoretical analysis to establish its equivalence and optimality. To align the agent's policy with this objective, we fine-tune the model using a Diplomacy-specific dataset structured in autoregressive form and a carefully designed loss function. This fine-tuning process enhances the agent's strategic capabilities, enabling it to outperform a diverse range of opponents, including state-of-the-art models such as Cicero, in Diplomacy.

Our contributions can be summarized as follows: 1) we introduce DipLLM, an LLM-based autoregressive factorization agent for Diplomacy that simplifies complex decision-making and fine-tuning. 2) we formally define the equilibrium policy within the factorization framework and fine-tune DipLLM to align with this objective using a specially designed loss function and a collected dataset. 3) we demonstrate that our agent outperforms previous domain-specific models, including Cicero and other LLM-based approaches, in no-press Diplomacy—achieving this with only $1.5\%$ of the data required by Cicero.

## 2. Related Work

**AI in Diplomacy.** Diplomacy has become a prominent benchmark for multi-agent games, known for its seven-

player mixed cooperative-competitive dynamics, simultaneous moves, and vast action space ranging from $10^{21}$ to $10^{64}$. The no-press variant of Diplomacy, which prohibits communication between players, demands implicit coordination through in-game actions, posing significant challenges for AI in strategic planning. Paquette et al. (2019b) introduced DipNet, the first deep learning-based Diplomacy agent, utilizing imitation learning on large-scale human gameplay data. Subsequent studies in no-press Diplomacy build upon similar architectures, incorporating reinforcement learning methods (Anthony et al., 2020; Bakhtin et al., 2021; 2022b). Recent advances predominantly rely on training agents using equilibrium search methods to approximate Nash equilibrium (Gray et al., 2020; Bakhtin et al., 2022a). While effective, these approaches require extensive computational resources to generate large-scale game data. For instance, Cicero utilizes the Coshar-piKL equilibrium search method to train a policy and value network, requiring up to 448 GPUs for gameplay rollouts. In contrast, our approach leverages the general reasoning capabilities of LLMs, enabling fine-tuning on a relatively small dataset while outperforming domain-specific models.

**LLM-based Agent.** LLM-based agent have gained generalized reasoning abilities from vast amounts of human language data, enabling their application to decision-making tasks. One line of research focuses prompting techniques (Sahoo et al., 2024) for enhancing the decision-making capabilities of large foundation models, with notable approaches including React (Yao et al., 2022), AutoGen (Wu et al., 2023), and Reflexion (Shinn et al., 2024). These prompt-based techniques have shown success in various scenarios, such as personal assistants (Li et al., 2024), robotics (Cheng et al., 2024; Chen et al., 2025a) and games (Gallotta et al., 2024; Xu et al., 2023; Guan et al., 2024). However, prompt-based techniques offer only short-term performance improvements and are inherently constrained by the capabilities of the underlying foundation models. Another line of research focuses on fine-tuning whole LLMs with domain-specific data to enhance their decision-making performance (Pan et al., 2024; Zhai et al., 2024). These fine-tuning approaches have achieved promising results, though primarily in relatively simple decision-making environments. Unlike these methods, we fine-tune LLMs specifically for complex multiplayer games, Diplomacy, enhancing their strategic decision-making capabilities.

## 3. Preliminary

### 3.1. Markov Games

We model Diplomacy as a multiplayer Markov game. Formally, an $n$-player Markov game $\Delta$ is defined as a tuple: $\Delta = \langle S, A_1, \ldots, A_n, r_1, \ldots, r_n, p \rangle$ where $S$ is the state space, $A_i$ is the action space of player $i$ ($i = 1, \ldots, n$),

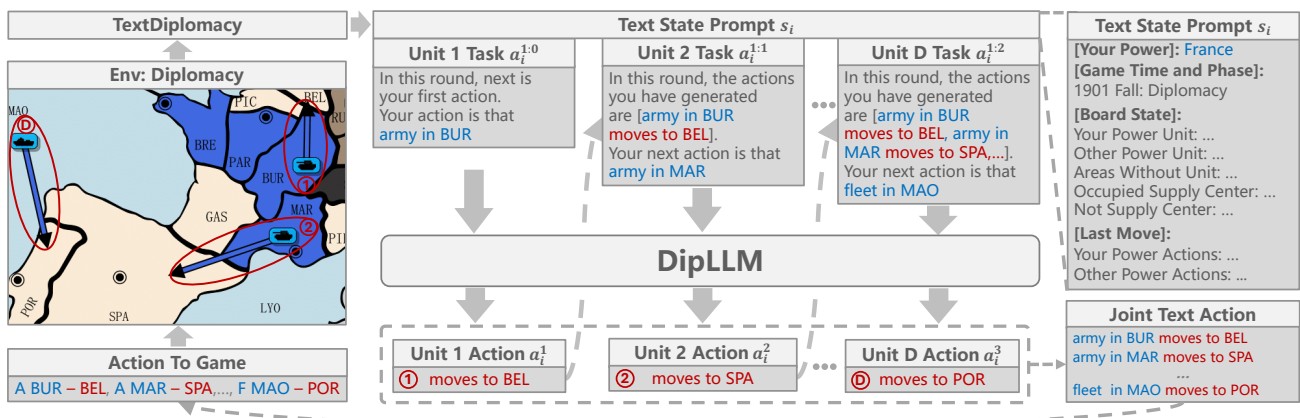

*Figure 2.* The inference process of our LLM-based autoregressive factorization agent. DipLLM sequentially decides the action for each unit by prompting the Text State and unit Task, and finally sends the joint actions to the environment for execution.

$r_i : S \times A_1 \times \cdots \times A_n \to \mathbb{R}$ is the reward function for player $i$, $p : S \times A_1 \times \cdots \times A_n \to S$ is the transition function. The objective of each player $i$ is to select a policy $\pi_i(s) : S \to \Delta A_i$ that maximizes their expected reward, given the policies of the other players. In each state $s$, every player $i$ simultaneously selects an action $a_i$ from their action set $A_i$. The actions of all other players, excluding $i$, are represented as $\boldsymbol{a}_{-i}$. Players may select a probability distribution over actions, where the probability of action $a_i$ is denoted $\pi_i(a_i|s)$.

### 3.2. Equilibrium Search

**piKL-Hedge** (Jacob et al., 2022) is a typical equilibrium search algorithm that iteratively converges to an equilibrium. The core idea behind piKL-Hedge is to apply a KL-divergence constraint to guide the policy search, ensuring that the resulting policies stay close to human strategies. In this approach, each player $i$ aims to maximize their expected reward while also remaining "close" to a fixed anchor policy $\tau_i$. When player $i$ selects action $a_i$ while other players choose actions $a_{-i}$, the utility $u_i(a_i, a_{-i})$ is computed using a pre-trained value function derived from reinforcement learning. The average reward for action $a_i$ up to iteration $t$ is given by: On each iteration $t$ of piKL-Hedge, the policy $\pi_i^t$ is set according to

$$\pi_i^t(a_i|s) \propto \exp\left\{\beta \log(\tau_i(a_i|s)) + Q_i^t(s, a_i)\right\} \quad (1)$$

Where $Q_i^t(s, a_i) = \mathbb{E}_{\boldsymbol{a}_{-i} \sim \boldsymbol{\pi}_{-i}^t(\cdot|s)}[u_i(s, a_i, \boldsymbol{a}_{-i})]$. The parameter $\beta$ controls the relative influence of the reward function and the anchor policy.

### 3.3. LLM-Based Decision-Making Policy

We define $\mathcal{V}$ as the finite and discrete token vocabulary, with $\mathcal{V}^m$ and $\mathcal{V}^n$ representing the input and output text

spaces, constrained by maximum token lengths $m$ and $n$, respectively (Zhai et al., 2024). The input text $\boldsymbol{v}^{\text{in}} \in \mathcal{V}^m$ represents the state $s \in S = \mathcal{V}^m$, while the output text $\boldsymbol{v}^{\text{out}} \in \mathcal{V}^n$ corresponds to the action $a \in A$. An LLM policy, parameterized by $\phi$, maps inputs to outputs as $\pi_\phi : \mathcal{V}^m \to \mathcal{V}^n$. The probability of generating an output $\boldsymbol{v}^{\text{out}}$ given an input $\boldsymbol{v}^{\text{in}}$ is denoted by $\pi_\phi(\boldsymbol{v}^{\text{out}}|\boldsymbol{v}^{\text{in}})$.

In the context of Diplomacy, the input $\boldsymbol{v}^{\text{in}}$ encodes the game state, and the output $\boldsymbol{v}_i^{\text{out}} = \pi_\phi(\boldsymbol{v}^{\text{in}})$ represents the LLM-generated action, which is parsed into specific moves for the player $i$'s units. By establishing the correspondence between $\boldsymbol{v}^{\text{in}}$ in and $s$, and $\boldsymbol{v}^{\text{out}}$ and $a_i$, the policy probability simplifies to $\pi_\phi(a_i|s)$, providing a direct mapping from game states to actions.

## 4. DipLLM: A Fine-tuned LLM-based Agent

DipLLM is an LLM-based agent fine-tuned to learn equilibrium policies for Diplomacy. To reduce the complexity of joint units decision-making, we introduce an autoregressive factorization framework. Within this framework, we define a learning objective that approximates the Nash equilibrium, and the LLM is fine-tuned on a collected dataset to align with this objective.

### 4.1. LLM-based Autoregressive Factorization

In Diplomacy, a player can control up to 34 units, each with approximately 26 possible actions resulting in exponential growth in possible action combination. To tackle such combinatorial challenges, many traditional methods employ factorization techniques (Farnoosh et al., 2021; Chebotar et al., 2023; Zhu et al., 2025). Building on this, we design an LLM-based autoregressive factorization framework, which decomposes the task of selecting joint actions into a series of smaller, sequential sub-tasks.

The policy $\boldsymbol{\pi}_i(a_i^{1:D}|s)$ in Diplomacy determines the joint action of all $D$ units controlled by player $i$ based on the board state $s$. We decompose this policy into a sequence of sub-policies $\pi_i^d$, where each sub-policy decides the action of a single unit $d$. Each sub-policy $\pi_i^d$ considers the board state $s$ and the sequence of actions chosen for previous units, denoted as $\pi_i^d(\cdot|s, a_i^1, \ldots, a_i^{d-1})$. This structure ensures that decisions for each unit depend on both the board state and the context established by earlier decisions. To align this formulation with the next-token prediction framework used in language models, the sequence of previously selected actions $\{a_i^1, \ldots, a_i^{d-1}\}$ is treated as the context $a_i^{1:d-1}$. Consequently, the probability of selecting $a_i^d$ is written as $\pi_i^d(a_i^d|s, a_i^{1:d-1})$. This factorization reduces the combinatorial action space into manageable sub-action spaces, effectively overcoming the challenges posed by complex environments like Diplomacy.

Building on this, we can now describe the inference process for the LLM agent within this autoregressive factorization framework. As shown in Figure 2, TextDiplomacy module first converts the raw board state from environment into a text-based representation $s$. This representation serves as the foundational context for the agent. For each unit, a task-specific prompt is generated that includes the actions of previous units, $a_i^{1:d-1}$, as well as the unit's identity. Using this combined prompt, the LLM generates the current unit's action $a_i^d$, such as *moves to POR*, following the policy $\pi_i^d(a_i^d|s, a_i^{1:d-1})$. After generating actions for all units, the post-processing module maps these actions to their corresponding units, forming the complete joint action $a_i^D$, e.g. *...fleet in MAO moves to POR*. This final joint action is then formatted according to the game rules and sent to the environment for execution e.g. *...F MAO – POR*.

### 4.2. Learning Objective in Autoregressive Factorization

LLM-based agents that are not fine-tuned for equilibrium strategies struggle to handle complex strategic interactions in Diplomacy. To guide the fine-tuning process, we first define a learning objective for the equilibrium policy within the autoregressive factorization framework. Inspired by the final iterative policy in piKL-Hedge (Eq.1), we rewrite it for clarity and then use this refined version to define the equilibrium policy in our autoregressive factorization framework.

$$\boldsymbol{\pi}_i^*(a_i^{1:D}|s) \propto \exp\left\{\beta \log(\boldsymbol{\tau}_i(a_i^{1:D}|s)) + \boldsymbol{Q}_i(s, a_i^{1:D})\right\} \quad (2)$$

where $a_i^{1:D}$ represents the joint action of $D$ units for player $i$, $\boldsymbol{\tau}_i$ serves as the human-imitative anchor policy, and $\boldsymbol{Q}_i$ represents the reward for the joint action.

To enable autoregressive factorization, we decompose the joint Q-value into a sequence of conditional Q-values, defining a separate Q-value $Q_i^d$ for each unit $d$. The Q-value

for each individual unit action $a_i^d$, conditioned on prior unit actions, is defined as:

$$Q_i^d(s, a_i^{1:d-1}, a_i^d) \triangleq \log \sum_{a_i^{d+1:D}} \exp\left\{\boldsymbol{Q}_i(s, a_i^{1:d}, a_i^{d+1:D})\right.$$
$$\left. + \beta \log \boldsymbol{\tau}_i(a_i^{1:d}, a_i^{d+1:D}|s)\right\} \quad (3)$$

Specifically, when $d < D$, $\sum_{a_i^{d+1:D}}$ denotes the summation over all possible combinations of subsequent actions $a_i^{d+1}, a_i^{d+2}, \ldots, a_i^D$. When $d = D$, this term is defined to be 1, since there are no remaining actions to marginalize. The $\beta$ controls the trade-off between the expected cumulative reward of action $a_i^d$ and anchor policy. Our defined unit-level state-action value function $Q_i^d\left(s, a_i^{1:d-1}, a_i^d\right)$ captures the trade-off between the expected cumulative reward of selecting action $a_i^d$ and the behavior under an anchor policy. Based on this formulation, the policy learning objective for autoregressive factorization is defined as:

$$\pi_i^{d,*}(a_i^d|s, a_i^{1:d-1}) \triangleq \frac{\exp\left\{Q_i^d(s, a_i^{1:d-1}, a_i^d)\right\}}{\sum_{a_i^d} \exp\left\{Q_i^d(s, a_i^{1:d-1}, a_i^d)\right\}} \quad (4)$$

To elucidate the properties of the proposed objective in game settings, we provide a theoretical analysis.

**Theorem 1** (Objective Equivalence). *The joint policy derived using the learning objective in autoregressive factorization form,* $\prod_{d=1}^D \pi_i^{d,*}(a_i^d|s, a_i^{1:d-1})$, *is equivalent to the original policy distribution* $\boldsymbol{\pi}_i^*$.

Theorem 1 demonstrates that the joint policy derived from autoregressive factorization is theoretically equivalent to the original policy. This equivalence offers a unit-based optimization approach, allowing fine-tuned autoregressive factorization agents to effectively approximate the original Nash equilibrium. By breaking down the complex action space, the learning objective ensures that the LLM makes decisions within a manageable sub-action space for each unit, while guaranteeing that the resulting joint policy remains equivalent to the original policy.

**Theorem 2** (Optimality of objective in 2p0s games). *In two-player zero-sum games, when both players update their policies using the learning objective in autoregressive factorization over $T$ iterations, their average policies converge to a $(max_{i=1,2}\beta_i\delta_i)$-approximate Nash equilibrium as $T \to \infty$. Here, $\delta_i$ is defined as* $\max_{a^{1:D}\sim\mathcal{A}_i} \log\left(1/\boldsymbol{\tau}_i\left(a^{1:D}\right)\right)$.

Theorem 2 establishes the optimality of the proposed learning objective in two-player zero-sum games. Extensive research (Bakhtin et al., 2021; 2022b) indicates that that theoretical insights from these settings can generalize effectively to multiplayer games. This supports the applicability of our approach to complex environments like Diplomacy. Full proofs are provided in the Appendix C.

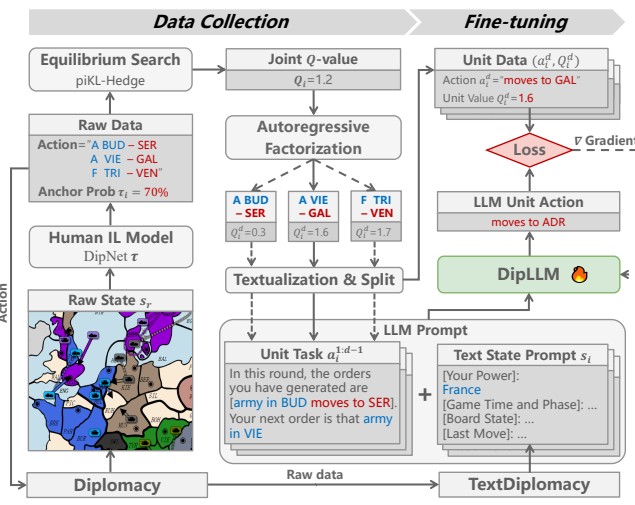

*Figure 3.* Pipeline for fine-tuning the LLM-based autoregressive factorization agent. Raw data is collected by interacting with the environment via DipNet. Q-values are generated through piKL-Hedge search, and the data is stored in prompt form. The LLM is then fine-tuned using a designed loss function.

### 4.3. Fine-tuning for Equilibrium Policy

The fine-tuning process, as depicted in Figure 3, involves optimizing a loss function using game-specific data structured in an autoregressive factorization format.

**Loss Function.** The objective of fine-tuning is to align the policy of LLM-based agent with the equilibrium policy defined by the optimization objective in Equation 4. This alignment is achieved by minimizing the KL divergence between the LLM's policy and the target equilibrium policy. Formally, the objective is expressed as:

$$\min_{\pi_\phi} \mathbb{E}_{(s, a_i^{1:d-1}) \sim \mathcal{D}} \left[ D_{\mathrm{KL}} \left( \pi_i^{d,*}(\cdot|s, a_i^{1:d-1}) \| \pi_\phi(\cdot|s, a_i^{1:d-1}) \right) \right] \tag{5}$$

Through derivation, we find that this objective is equivalent to maximizing:

$$\max_{\pi_\phi} \mathbb{E}_{(s, a_i^{1:d-1}, a_i^d) \sim \mathcal{D}} \big[ \log \pi_\phi \left( a_i^d | s, a_i^{1:d-1} \right) \\ \cdot \exp \left\{ Q_i^d(s, a_i^{1:d-1}, a_i^d) \right\} \big] \tag{6}$$

This loss function comprises two key components. The term $\log \pi_\phi(a_i^d|s, a_i^{1:d-1})$ corresponds to the supervised fine-tuning (SFT) component, aligning the LLM's output actions with those in the collected dataset. The remaining terms act as weights, ensuring that the learned policy approximates the equilibrium policy by incorporating the unit value estimates $Q_i$.

**Data Collection.** To fine-tune our LLM-based autoregressive factorization agent, we collect raw data through interactions between the domain-specific model DipNet (Paquette

et al., 2019a) and the Diplomacy environment. This initial dataset forms the foundation for subsequent training. Next, we apply the equilibrium search algorithm, piKL-Hedge, using DipNet as a human-like anchor policy $\boldsymbol{\tau}_i$. This process generates joint action Q-values $\boldsymbol{Q}_i$, which represent their expected rewards.

To enable autoregressive policy training, we decompose the joint Q-values into unit-level training samples, denoted as $Q_i^d$. Directly computing the Q-value defined in Equation 3 requires applying the log-sum-exp function over the entire action space of $a_i^d$, which is computationally expensive and inefficient. To improve efficiency, we approximate this Q-value by using a tight lower bound based on the sampled values of $\boldsymbol{Q}_i(s, a_i^{1:D}) + \beta \log \boldsymbol{\tau}_i(a_i^{1:D}|s))$, where $a_i^{1:D} \sim \boldsymbol{\pi}_i^*(\cdot|s)$. The derivation of this approximation is provided in the Appendix C. After compute unit Q-value, we convert actions into text format and split them to extract unit-specific task prompts $a_i^{1:d-1}$ and corresponding ground truth actions $a_i^d$. The unit-specific task prompts $a_i^d$ are then combined with the text board state $s$ from TextDiplomacy to serve as the LLM's input. Finally, the processed data is stored as autoregressive factorization transitions in the form $(s, a_i^{1:d-1}, a_i^d, Q_i^d)$, where:

- $s$: a textual board state for player $i$,
- $a_i^{1:d-1}$: a task prompt containing the actions of the previous d-1 units,
- $a_i^d$: a textual representation of the ground truth action for the $d$-th unit,
- $Q_i^d$: the Q value corresponding to the $d$-th unit's action

## 5. Experiments

We conduct a comprehensive evaluation of DipLLM across various scenarios to assess its effectiveness. First, we evaluate its performance against a pool of baseline opponents, including the SOTA Cicero, with all agents operating without equilibrium search techniques for a more time-efficient assessment. We further compare DipLLM with the enhanced agent, Cicero with equilibrium Search, for a deeper evaluation of its strategic decision-making capabilities. Additionally, we examine the benefits of fine-tuning by comparing the performance of the fine-tuned LLM agent with that of a domain-specific model, highlighting the advantages of our approach. Finally, we conduct ablation studies to analyze the contributions of the autoregressive factorization and the fine-tuning process.

### 5.1. Experimental Setup

**Evaluation Method.** We evaluate DipLLM by conducting numerous no-press Diplomacy games against baseline models. In each game, one agent controls a single power, while the other six powers are controlled by copies of the opposing

| Agent | SoS Scores↑ | Win Rate↑ | Most SC↑ | Survived↑ | Defeated↓ |
|---|---|---|---|---|---|
| **DipLLM (Ours)** | **23.0%**±0.1% | **22.3%**±0.2% | **29.3%**±0.2% | **50.3%**±0.7% | **27.4%**±0.1% |
| Cicero (Bakhtin et al., 2022a) | 20.8%±0.3% | 20.5%±0.2% | 28.7%±0.2% | 50.1%±0.5% | 29.4%±0.2% |
| DNVI (Bakhtin et al., 2021) | 6.6%±0.1% | 4.3%±0.1% | 5.0%±0.1% | 35.8%±0.3% | 59.9%±0.7% |
| DORA (Bakhtin et al., 2021) | 5.7%±0.1% | 4.7%±0.1% | 3.9%±0.2% | 33.0%±0.6% | 62.3%±0.8% |
| DipNet (Anthony et al., 2020) | 1.8%±0.1% | 1.1%±0.1% | 1.1%±0.1% | 30.1%±0.4% | 68.8%±0.7% |

*Table 1.* Performance of different agents in a population of various agents. The ± indicates one standard error.

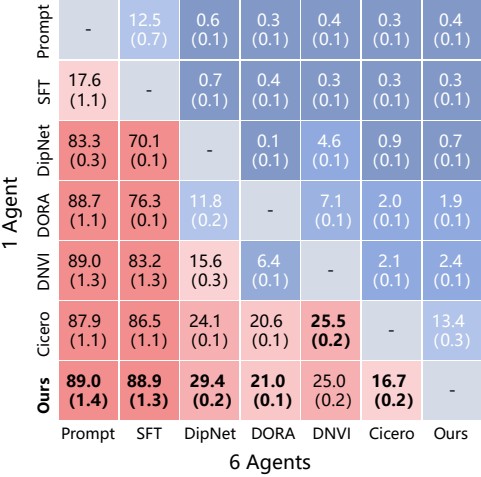

*Figure 4.* SoS scores of various agents (y-axis) when competing against six identical copies of another agent (x-axis). The values in parentheses represent one standard error. Note that equal performance corresponds to $1/7 \approx 14.3\%$.

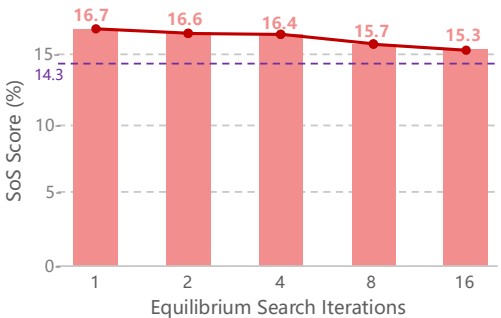

*Figure 5.* Performance comparison of our agent playing against six Cicero agents, with each Cicero agent incorporating different numbers of equilibrium search iterations. The purple dashed line indicates equal performance, i.e., a win rate of $1/7 \approx 14.3\%$.

agent, forming a 1v6 competition (Bakhtin et al., 2021). For computational efficiency, agents directly output their actions during the inference phase without employing additional equilibrium search. This approach is practical, as equilibrium search modules are typically resource-intensive and time-consuming. Importantly, this approach ensures a fair evaluation, as all models have the potential to incorporate search-based enhancements in future deployments.

**Evaluation Metrics.**

We evaluate the agents using two key metrics. The first is the **sum-of-squares scoring (SoS)**, which is commonly used in previous studies (Bakhtin et al., 2022b;a). In this metric, player $i$ receives a score of $\frac{C_i^2}{\sum_{j \in N} C_j^2}$, where $C_i$ represents the number of supply centers (SCs) controlled by player $i$. The average score for an identical agent is $1/7 \approx 14.3\%$ The second metric is based on the four possible game outcomes: **win**, **most SCs**, **survived**, and **defeated**. A power wins by controlling 18 or more of the 34 SCs, ending the game. A power is defeated if it loses all its SCs. If no power wins, the game ends in a draw, with the power controlling the most SCs labeled as having the most SCs. When the game ends,

any power without SCs is considered defeated, while powers still controlling at least one SC are considered survived.

### 5.2. Experimental Results

**Baseline Agents** We evaluate DipLLM's performance against five open-source domain-specific models and two LLM-based decision-making agents employing either prompting or supervised fine-tuning (SFT) techniques:

- **DipNet** (Anthony et al., 2020): A 0.3B parameter model trained via imitation learning on a large-scale dataset of human expert demonstrations. It serves as a baseline agent for comparison and is also used to interact with the environment for raw data collection.
- **DORA** (Bakhtin et al., 2021): Built on the DipNet architecture, DORA incorporates an equilibrium search module to train both policy and value networks from scratch using reinforcement learning.
- **DNVI** (Bakhtin et al., 2021): Similar to DORA but with policy and value networks initialized from human behavior cloning (BC) pretraining.
- **Cicero** (Bakhtin et al., 2022a): The SOTA model with 2.7 billion parameters, trained using large-scale human demonstrations and the equilibrium search technique CoShar-piKL.
- **Prompt**: A pure prompt-based LLM agent developed with techniques such as social reasoning, reflection and subgoal generation, as described in Richelieu (Guan

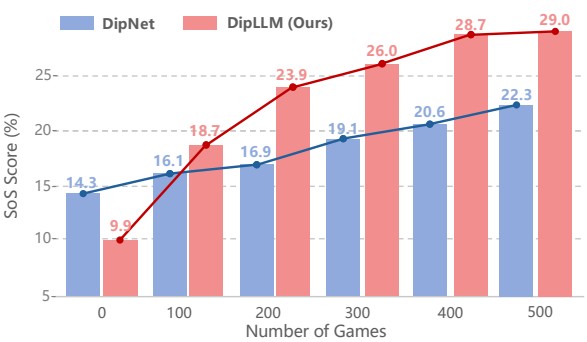

*Figure 6.* Performance comparison of fine-tuned DipLLM and Dip-Net (Paquette et al., 2019a), both trained using different numbers of games. Each model is evaluated by playing against six original DipNet agents without fine-tuning.

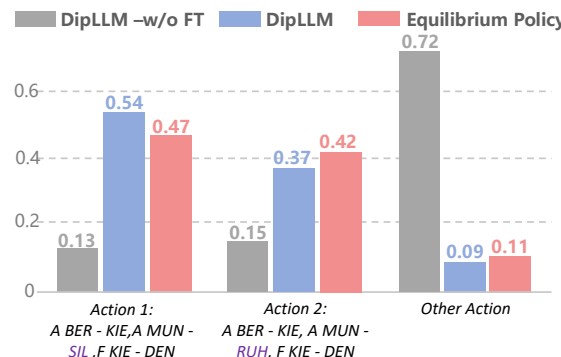

*Figure 7.* Comparison of action distributions between DipLLM, DipLLM (w/o fine-tuning, FT), and the equilibrium policy generated by piKL-Hedge.

et al., 2024).

- **SFT**: An LLM agent trained with SFT on the same data, without using autoregressive factorization, generating all unit actions in a single step.

**Evaluation in Baseline Population.** To accurately evaluate the equilibrium policy of our agent, we construct a population comprising DipLLM and the diverse baseline opponents. Games are then conducted in a 1v6 competition format, with agents randomly selected from this population. Figure 4 illustrates the SoS scores from pairwise agent competitions. The results reveal that both the prompt-based and SFT agents underperform. To maintain the competitiveness of the opponent population, we exclude these two models and provide a more detailed evaluation of the performance across the remaining models. Table 1 presents the performance results for the agents in this population. DipLLM outperforms all other baselines across every metric, with a 2.2% advantage in the SoS score and a 1.8% higher win rate compared to the second-best model. This demonstrates its strong decision-making capabilities. Additionally, while DipLLM's survival rate is almost identical to Cicero's, its significantly higher win rate and reduced failure rate suggest a more aggressive and effective decision-making approach.

**Playing against Cicero including Equilibrium Search Module.** To further evaluate DipLLM's decision-making capabilities, we directly pit it against Cicero, which utilizes an equilibrium search module with varying rollout counts to enhance its strategic depth. Although DipLLM is also compatible with the equilibrium search module, this analysis focuses on its end-to-end reasoning performance without relying on additional search steps. Figure 5 shows that, even as the number of rollouts increases and strengthens Cicero's equilibrium search, DipLLM continues to outperform it, maintaining a win rate above the average (14.3%). This demonstrates DipLLM's stable and robust decision-making performance. To further explore DipLLM's potential when

combined with equilibrium search, we conducted a preliminary experiment under limited computational resources. The results are included in Appendix E.2. In addition to its strong performance, DipLLM demonstrates significantly improved inference efficiency. It requires only 10–20% of Cicero's inference time per turn, making it a more practical choice in real-time decision-making scenarios.

**Comparison of Fine-tuning DipLLM to Domain Model DipNet.** We compare the performance of DipLLM, a large pre-trained language model, with DipNet, a domain-specific model, after fine-tuning on the same dataset divided into subsets of varying sizes. DipNet is fine-tuned on the collected raw data following the approach in (Bakhtin et al., 2021), while DipLLM is fine-tuned using our proposed loss function on text data. Figure 6 highlights the consistent performance improvements of both models as the dataset size increased. Initially, DipNet, a task-specific model pretrained on domain-relevant data, outperforms DipLLM due to its specialized architecture and prior domain knowledge. In contrast, DipLLM, which lacks domain-specific pretraining, starts with lower performance. However, DipLLM demonstrates remarkable data efficiency during fine-tuning. With only 100 games of fine-tuning data, DipLLM not only matches but surpasses DipNet's performance. As the dataset size grows to 500 games, DipLLM achieves a significant lead, outperforming DipNet by 6.7%. These results highlight DipLLM's ability to leverage its general reasoning and decision-making capabilities, achieving superior performance with relatively small fine-tuning data. This comparison underscores the effectiveness of our approach and the advantages of integrating large pre-trained language models into domain-specific tasks.

**Ablation Study.** This ablation study investigates the critical roles of autoregressive factorization (AF) and fine-tuning in optimizing DipLLM's performance. The *Without AF* agent uses an LLM to select actions for all units in a single step

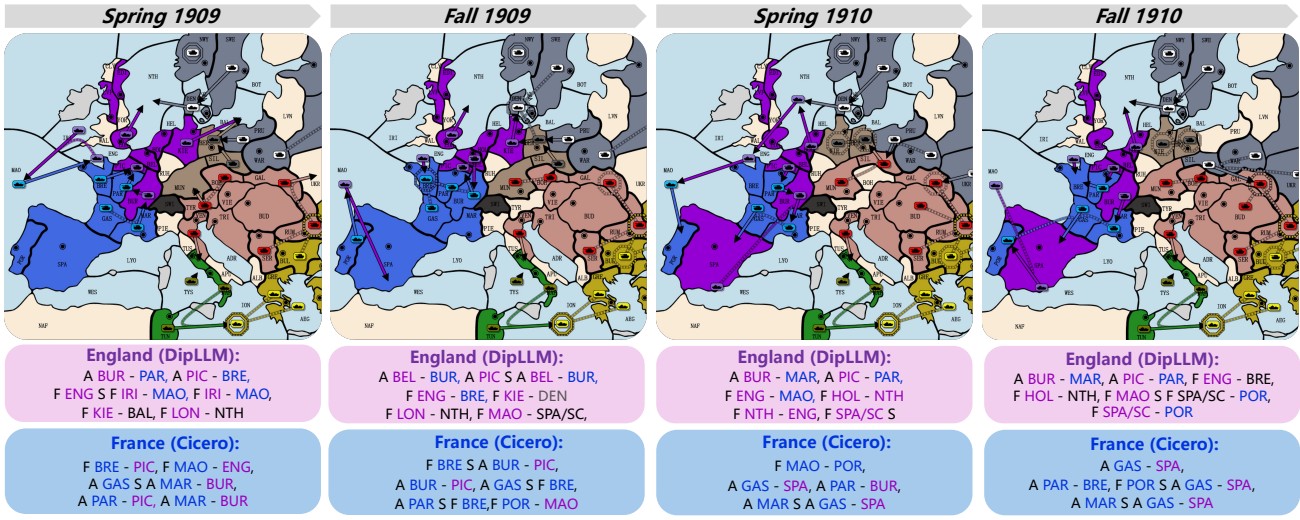

*Figure 8.* An example of DipLLM-controlled England strategically defeating Cicero-controlled France.

| AF | Fine-tune | SoS Score ↑ | Win ↑ | Survived ↑ | Defeated ↓ |
|----|-----------|-------------|-------|------------|------------|
| ✗ | ✗ | 0.2% | 0.0% | 4.3% | 95.7% |
| ✗ | ✓ | 0.8% | 0.0% | 19.2% | 80.8% |
| ✓ | ✗ | 9.9% | 6.7% | 40.0% | 53.3% |
| ✓ | ✓ | **29.4%** | **25.2%** | **45.9%** | **29.0%** |

*Table 2.* Ablation study on the effects of autoregressive factorization and fine-tuning when playing against six DipNet agents.

(one-step decision-making). The *Only Fine-tune* method fine-tunes the one-step decision-making LLM using joint action datasets. As shown in Table 2, fine-tuning alone yields minimal gains, as the one-step decision-making framework still struggles with the vast action space. Incorporating AF mitigates this challenge by decomposing the combinatorial action space into smaller sub-action spaces, resulting in improved performance. However, since the AF-based agent is not fine-tuned for an equilibrium strategy, its performance remains suboptimal. When AF is combined with fine-tuning within the reduced sub-action spaces, performance improves significantly. This is because fine-tuning within smaller, well-defined sub-action spaces enables the model to adapt more effectively to the task. These results highlight the critical role of both AF and fine-tuning in addressing complex action spaces and optimizing LLM-based agents for strategic games.

To further assess whether fine-tuning aligns the agent with the equilibrium policy, we analyze the changes in the opening-turn action distributions for England, as controlled by DipLLM. Specifically, we compare the agent's action distributions with the equilibrium policy derived from the piKL-Hedge algorithm. As shown in Figure 7, England's

equilibrium strategy in the opening phase primarily involves two actions: moving the MUN army to either SIL (Action 1) or RUH (Action 2). Before fine-tuning, the agent assigns low probability to these critical actions, favoring suboptimal alternatives. After fine-tuning, the agent's action distribution shifts to more closely align with the equilibrium policy, with significantly higher probabilities for Actions 1 and 2. This shift demonstrates that fine-tuning effectively guides the agent toward equilibrium-like strategies.

**A Case Study: Feint to Attack, Strike Where Least Expected.** DipLLM demonstrates exceptional strategic decision-making, using misdirection to exploit opponent vulnerabilities. As shown in Figure 8, in Spring 1909, England (DipLLM) faced a tense standoff with France (Cicero) on the western front while under pressure from Russia and Germany. To break the deadlock, DipLLM staged a diversion with armies in BUR and PIC to tie down France, while fleets in IRI and ENG launched a surprise attack on the poorly defended MAO region. This feint distracted French forces in BRE and GAS, forcing them to retreat to POR. England capitalized by capturing SPA by Fall 1910, bypassing French forces in POR and effectively encircling France. A few turns later, England conquered all French territories, securing a decisive victory. This case highlights DipLLM's effective use of deception and multi-front strategy to turn a vulnerable position into a comprehensive triumph.

## 6. Conclusions

In this work, we propose DipLLM, a fine-tuned LLM-based agent that learns equilibrium policies for Diplomacy. DipLLM addresses the complexity of decision-making by leveraging autoregressive factorization to decompose tasks

into smaller, sequential steps. Building on this foundation, we define a theoretically grounded learning objective to approximate the Nash equilibrium and fine-tune DipLLM to align with this objective. Our agent surpasses SOTA Cicero while using only $1.5\%$ of Cicero's training data. These results highlight the potential of LLM-based agents for addressing complex decision-making challenges and pave the way for their broader applications in the future.

**Limitations**. While fine-tuning DipLLM still relies on external data generated by other models through equilibrium search, generating data via self-play with the agent offers a more scalable solution for broader application scenarios. We provide preliminary evidence that combining LLM-based agents with real-time equilibrium search techniques can significantly enhance decision-making performance. However, due to computational constraints, we were unable to conduct a more extensive investigation of this approach.

## Acknowledgements

This work was supported in part by the National Natural Science Foundation of China under Grant 62293541 and Grant 62136008, in part by Beijing Natural Science Foundation under Grant 4232056, and in part by Beijing Nova Program under Grant 20240484514, in part by the Strategic Priority Research Program of Chinese Academy of Sciences under Grant No. XDA27030100. We would also like to thank Zijie Zhao for his insightful discussions and valuable suggestions during the early stage of this work.

## Impact Statement

This paper presents work whose goal is to advance the field of Machine Learning. There are many potential societal consequences of our work, none which we feel must be specifically highlighted here.

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

# A. Description of Diplomacy

Diplomacy is a strategic board game simulating early 20th-century European geopolitics, where seven powers (Austria, England, France, Germany, Italy, Russia, and Turkey) compete to control 34 supply centers distributed across 75 provinces, including land and maritime territories. Gameplay progresses through annual cycles starting in 1901, each comprising five phases: Spring Movement, Spring Retreat, Fall Movement, Fall Retreat, and Winter Adjustment.

**Movement Phases**. During movement phases, players issue tactical commands to military units. A *hold* order maintains a unit's defensive position (default if no order is given). A *move* order directs a unit to attack adjacent provinces—armies traverse land or coastal regions, while fleets navigate coastal or aquatic territories. *Support* orders enhance allied attack/defense capabilities but require the supporting unit to legally access the target province (e.g., Marseille supporting Paris's advance to Burgundy); such support fails if the supporter is attacked. *Convoy* orders enable armies to cross water via coordinated fleet movements, requiring uninterrupted convoy paths and synchronized orders.

**Retreat Phases**. Displaced units from contested provinces must retreat or disband. Valid retreats target adjacent unoccupied provinces not involved in prior conflicts. Disbanding occurs automatically if retreat destinations are unavailable, conflicting, or if orders are omitted.

**Adjustment Phases**. Annual adjustments align unit counts with controlled supply centers. Powers exceeding their supply center count disband surplus units, while those with excess centers build new units in unoccupied original territories (e.g., Germany builds in Berlin if controlled). Players may voluntarily waive build rights.

**Tacit Communication in No-Press Diplomacy**. Despite explicit communication restrictions, players convey intentions implicitly through tactical orders. Offensive unit positioning signals aggression, support orders imply alliances, and convoy proposals suggest cooperation. Even invalid orders—such as Russia supporting an impossible English attack from Paris to London—can strategically influence opponents by signaling priorities. This emergent negotiation layer enriches gameplay through action-based inference rather than verbal coordination.

The game concludes when a single power secures majority control of supply centers, demanding strategic foresight, adaptive coordination, and precise interpretation of opponents' tactical signals.

# B. Full Prompts for DipLLM

We utilize the Llama 3 8B model as the backbone of our framework. The required training data, represented in the form $(s, a_i^{1:d-1}, a_i^d, Q_i^d)$, is structured and stored as part of the prompt.

## B.1. System Prompt

The system prompt used in our method is listed below.

> {
> "role": "system",
> "content": "You are an expert in the no-press Diplomacy game environment. As one of seven powers, your task is to use your army and fleet to control the supply centers on the board. You are playing [Your Power] and observing [Game Time and Phase], [Board State], and [Last Moves] below. In the [Board State], each power will sequentially display the locations of its army and fleet. Remember, unless specified otherwise, we will omit the default attributes for areas, which include the coast, neither supply center nor home center, no troops dislodged, and not occupied by anyone."
> }

## B.2. User Prompt

The user prompt, which includes the components $s$ and $a_i^{1:d-1}$, is outlined below as utilized in our methodology.

{
"role": "user",
"content": "
*### textual board state s*
**[Game Time and Phase]:**
1905 Spring: Diplomacy
**[Board State]:**
**Your Power Unit:**
Turkey's army: Bulgaria (including Eastern and South Coast, Turkey's supply center)
Turkey's fleet: Aegean Sea (water), Black Sea (water), Ionian Sea (water)
Turkey's center without units: Ankara, Constantinople, Smyrna
**Other Power Unit:**
Austria's army: Budapest (land, Austria's supply center), Greece (Austria's supply center), Serbia (land, Austria's supply center), Tyrol (land), Venice (Austria's supply center)
Austria's fleet: Trieste (Austria's supply center)
Austria's center without units: Vienna
England's army: Denmark (England's supply center)
England's fleet: Norway (England's supply center)
England's center without units: Sweden
France's army: Burgundy (land), Gascony, Marseilles (France's supply center), Picardy, York
France's fleet: English Channel (water), Heligoland Bight (water), London (France's supply center)
France's center without units: Brest, Edinburgh, Liverpool, Paris, Portugal, Spain
Germany's army: Belgium (Germany's supply center), Holland (Germany's supply center), Ruhr (land)
Germany's fleet: Kiel (Germany's supply center), North Sea (water)
Germany's center without units: Berlin, Munich
Italy's army: Piedmont, Rome (Italy's supply center)
Italy's fleet: Tunisia (Italy's supply center)
Italy's center without units: Naples
Russia's army: Moscow (land, Russia's supply center), Rumania (Russia's supply center), Warsaw (land, Russia's supply center)
Russia's fleet: Finland, Sevastopol (Russia's supply center)
Russia's center without units: St. Petersburg
**Areas Without Unit:**
unoccupied supply center: None
occupied supply center: Ankara (Turkey's), Berlin (Germany's), Brest (France's), Constantinople (Turkey's), Edinburgh (France's), Liverpool (France's), Munich (land, Germany's), Naples (Italy's), Paris (land, France's), Portugal (France's), Smyrna (Turkey's), Spain (including North and South Coast, France's), St. Petersburg (including North and South Coast, Russia's), Sweden (England's), Vienna (land, Austria's)
not supply center: Adriatic Sea (water), Albania, Apulia, Armenia, Baltic Sea (water), Barents Sea (water), Bohemia (land), Bothnia (water), Clyde, Eastern Mediterranean (water), Galicia (land), Irish Sea (water), Livonia, Lyon (water), Mid Atlantic Ocean (water), North Africa, North Atlantic Ocean (water), Norwegian Sea (water), Prussia, Silesia (land), Skagerrak (water), Syria, Tuscany, Tyrrhenian Sea (water), Ukraine (land), Wales, Western Mediterranean (water)
. . .

**[Last Move]:**
**Your Power Order:**
Turkey: army in Bulgaria supports fleet in Ionian Sea move to Greece, fleet in Aegean Sea moves to Smyrna, fleet in Black Sea supports army in Bulgaria, fleet in Ionian Sea moves to Greece
**Other Power Order:**
Austria: army in Galicia moves to Budapest, army in Greece supports army in Bulgaria, army in Serbia supports army in Greece, army in Trieste moves to Venice, army in Vienna moves to Tyrol
England: army in Denmark moves to Sweden, fleet in Norway moves to St. Petersburg's North Coast
France: army in Burgundy holds, army in Gascony supports army in Burgundy, army in Marseilles supports army in Burgundy, army in Paris moves to Picardy, army in York supports fleet in English Channel move to London, fleet in Brest moves to English Channel, fleet in English Channel moves to London, fleet in North Sea moves to Heligoland Bight
Germany: army in Belgium supports army in Munich move to Burgundy, army in Holland supports army in Belgium, army in Munich moves to Burgundy, army in Ruhr supports army in Belgium, fleet in Kiel moves to Denmark, fleet in London holds
Italy: army in Piedmont moves to Marseilles, army in Venice moves to Rome, fleet in Eastern Mediterranean moves to Smyrna, fleet in Tyrrhenian Sea moves to Tunisia
Russia: army in Moscow moves to St. Petersburg, army in Rumania supports army in Bulgaria, army in Ukraine moves to Warsaw, fleet in Finland moves to Sweden, fleet in Sevastopol supports army in Rumania

*### task prompt $a_i^{1:d-1}$*
In this round, the orders you have previously generated are [army in Bulgaria supports fleet in Ionian Sea move to Greece, fleet in Aegean Sea supports fleet in Ionian Sea move to Greece, fleet in Black Sea supports army in Bulgaria]. The candidate orders for fleet in Ionian Sea are [moves to Adriatic Sea, moves to Greece, moves to Naples, moves to Tunisia, supports army in Bulgaria move to Greece, supports fleet in Aegean Sea move to Greece]. The best order from candidate orders is that fleet in Ionian Sea"
}

## B.3. Other Prompts

The remaining prompts, which incorporate the ground truth data $a_i^d$ and $Q_i^d$, are provided below.

{
*### Ground truth action $a_i^d$*
"role": "assistant",
"content": "moves to Greece",
}
{
*### Unit Q-value $Q_i^d$*
"role": "value",
"content": "1.28",
}

## C. Theoretical Analysis and Proofs

First, we recall the definitions of factored Q-value and its corrsponding policy.

$$Q_i^d(s, a_i^{1:d-1}, a_i^d) \triangleq \log \sum_{a_i^{d+1:D}} \exp\left\{\boldsymbol{Q}_i(s, a_i^{1:d}, a_i^{d+1:D}) + \beta \log \boldsymbol{\tau}_i(a_i^{1:d}, a_i^{d+1:D}|s)\right\}$$

$$\pi_i^{d,*}(Q_i^d|s, a_i^{1:d-1}) \triangleq \frac{\exp\left\{Q_i^d(s, a_i^{1:d-1}, a_i^d)\right\}}{\sum_{a_i^d} \exp\left\{Q_i^d\left(s, a_i^{1:d-1}, a_i^d\right)\right\}} \tag{7}$$

**Theorem 1** (Objective Equivalence). *The joint policy derived using the learning objective in autoregressive factorization form, $\prod_{d=1}^{D} \pi_i^{d,*}(Q_i^d|s, a_i^{1:d-1})$, is equivalent to the original policy distribution $\boldsymbol{\pi}_i^*$.*

*Proof.* We establish equivalence through direct algebraic manipulation. Recalling the original policy definition in piKL-Hedge:

$$\boldsymbol{\pi}_i^*(a_i^{1:D}|s) \propto \exp\left\{\beta \log(\boldsymbol{\tau}_i(a_i^{1:D}|s)) + \boldsymbol{Q}_i(s, a_i^{1:D})\right\}$$

$$= \frac{\exp\left\{\beta \log(\boldsymbol{\tau}_i(a_i^{1:D}|s)) + \boldsymbol{Q}_i(s, a_i^{1:D})\right\}}{\sum_{a_i^{1:D}}\left[\exp\left\{\beta \log(\boldsymbol{\tau}_i(a_i^{1:D}|s)) + \boldsymbol{Q}_i(s, a_i^{1:D})\right\}\right]}$$

The factorization equivalence follows from:

$$\prod_{d=1}^{D} \pi_i^{d,*}(Q_i^d \mid s, a_i^{1:d-1}) = \prod_{d=1}^{D} \frac{\exp\left\{Q_i^d(s, a_i^{1:d-1}, a_i^d)\right\}}{\sum_{a_i^d} \exp\left\{Q_i^d(s, a_i^{1:d-1}, a_i^d)\right\}}$$

$$= \prod_{d=1}^{D} \frac{\exp\left\{\log \sum_{a_i^{d+1:D}} \exp\left\{\boldsymbol{Q}_i(s, a_i^{1:d}, a_i^{d+1:D}) + \beta \log \boldsymbol{\tau}_i(a_i^{1:d}, a_i^{d+1:D} \mid s)\right\}\right\}}{\sum_{a_i^d} \exp\left\{\log \sum_{a_i^{d+1:D}} \exp\left\{\boldsymbol{Q}_i(s, a_i^{1:d-1}, a_i^d, a_i^{d+1:D}) + \beta \log \boldsymbol{\tau}_i(a_i^{1:d-1}, a_i^d, a_i^{d+1:D} \mid s)\right\}\right\}}$$

$$= \prod_{d=1}^{D} \frac{\sum_{a_i^{d+1:D}} \exp\left\{\boldsymbol{Q}_i(s, a_i^{1:d}, a_i^{d+1:D}) + \beta \log \boldsymbol{\tau}_i(a_i^{1:d}, a_i^{d+1:D} \mid s)\right\}}{\sum_{a_i^{d:D}} \exp\left\{\boldsymbol{Q}_i(s, a_i^{1:d-1}, a_i^{d:D}) + \beta \log \boldsymbol{\tau}_i(a_i^{1:d-1}, a_i^{d:D} \mid s)\right\}}$$

$$= \frac{\exp\left\{\boldsymbol{Q}_i(s, a_i^{1:D}) + \beta \log \boldsymbol{\tau}_i(a_i^{1:D} \mid s)\right\}}{\sum_{a_i^{1:D}} \exp\left\{\boldsymbol{Q}_i(s, a_i^{1:D}) + \beta \log \boldsymbol{\tau}_i(a_i^{1:D} \mid s)\right\}}$$

$$= \boldsymbol{\pi}_i^*(a_i^{1:D} \mid s)$$

**Theorem 2** (Optimality of objective in 2p0s games). *In two-player zero-sum games, when both players update their policies using the learning objective in autoregressive factorization over $T$ iterations, their average policies converge to a $(max_{i=1,2}\beta_i\delta_i)$-approximate Nash equilibrium as $T \to \infty$. Here, $\delta_i$ is defined as $\max_{a^{1:D} \sim \mathcal{A}_i} \log\left(1/\boldsymbol{\tau}_i\left(a^{1:D}\right)\right)$.*

We extend the foundational result of Jacob et al. (2022) with the following advancements:

**Corollary 1** (Jacob et al., 2022). *For piKL-Hedge in two-player zero-sum games with regularized utilities $\mathcal{U}_i(\boldsymbol{\pi}_i) := \mathcal{U}_i(\boldsymbol{\pi}_i, \boldsymbol{a}_{-i}) = u_i(\boldsymbol{\pi}_i, \boldsymbol{a}_{-i}) - \beta_i D_{\mathrm{KL}}(\boldsymbol{\pi}_i \parallel \boldsymbol{\tau}_i)$, when both players update their policies using over $T$ iterations, their average policies converge to a Nash equilibrium:*

$$0 = \max_{\boldsymbol{\pi}_1^* \in \Delta(A_1)} \{\mathcal{U}_1(\boldsymbol{\pi}_1^*, \bar{\boldsymbol{\pi}}_2) - \mathcal{U}_1(\bar{\boldsymbol{\pi}}_1, \bar{\boldsymbol{\pi}}_2)\}$$

Building on this, we analyze the autoregressive factorization $\boldsymbol{\pi}_i^D := \prod_{d=1}^{D} \pi_i^{d,*}(\cdot|s, a_i^{1:d-1})$. Let $(\bar{\boldsymbol{\pi}}_1^D, \bar{\boldsymbol{\pi}}_2^D)$ denote any limit point of average policies. Theorem 1 ensures policy equivalence at each iteration, yielding:

$$
\begin{aligned}
0 = &\max_{\boldsymbol{\pi}_1^{D,*}\in\Delta(A_1)} \{\mathcal{U}_1(\boldsymbol{\pi}_1^{D,*},\bar{\boldsymbol{\pi}}_2^D) - \mathcal{U}_1(\bar{\boldsymbol{\pi}}_1^D,\bar{\boldsymbol{\pi}}_2^D)\} \\
= &\max_{\boldsymbol{\pi}_1^*\in\Delta(A_1)} \{u_1(\boldsymbol{\pi}_1^{D,*},\bar{\boldsymbol{\pi}}_2^D) - \lambda_1 D_{\mathrm{KL}}(\boldsymbol{\pi}_1^{D,*}\parallel\boldsymbol{\tau}_1) - u_1(\bar{\boldsymbol{\pi}}_1^D,\bar{\boldsymbol{\pi}}_2^D) + \lambda_1 D_{\mathrm{KL}}\left(\bar{\boldsymbol{\pi}}_1^D\parallel\boldsymbol{\tau}_1\right)\} \\
\geq &\max_{\boldsymbol{\pi}_1^{D,*}\in\Delta(A_1)} \{u_1\left(\boldsymbol{\pi}_1^{D,*},\bar{\boldsymbol{\pi}}_2^D\right) - u_1\left(\bar{\boldsymbol{\pi}}_1^D,\bar{\boldsymbol{\pi}}_2^D\right)\} - \lambda_1 D_{\mathrm{KL}}\left(\boldsymbol{\pi}_1^{D,*}\parallel\boldsymbol{\tau}_1\right) \\
= &\max_{\boldsymbol{\pi}_1^{D,*}\in\Delta(A_1)} \{u_1(\boldsymbol{\pi}_1^{D,*},\bar{\boldsymbol{\pi}}_2^D) - u_1(\bar{\boldsymbol{\pi}}_1^D,\bar{\boldsymbol{\pi}}_2^D)\} - \lambda_1 D_{\mathrm{KL}}\left(\prod_{d=1}^{D}\pi_1^{d,*}\left(\cdot|a_1^{1:d-1}\right)\Bigg\|\boldsymbol{\tau}_1(\cdot)\right) \\
= &\max_{\pi_1^*\in\Delta(A_1)} \Bigg\{u_1(\boldsymbol{\pi}_1^{D,*},\bar{\boldsymbol{\pi}}_2^D) - u_1(\bar{\boldsymbol{\pi}}_1^D,\bar{\boldsymbol{\pi}}_2^D) \\
&-\delta_1 \sum_{a_1^{1:D}\in\mathcal{A}_1^P}\left[\prod_{d=1}^{D}\pi_1^*\left(a_1^d|a_1^{1:d-1}\right)\log\left(\prod_{d=1}^{D}\pi_1^*\left(a_1^d|a_1^{1:d-1}\right)\right)\right] \\
&-\delta_1 \sum_{a_1^{1:D}\in\mathcal{A}_1^P}\left[\prod_{d=1}^{D}\pi_1^{d,*}\left(a_1^d|a_1^{1:d-1}\right)\cdot\log\left(1/\boldsymbol{\tau}_1(a_1^{1:D})\right)\right]\Bigg\} \\
\geq &\max_{\pi_1^*\in\Delta(A_1)} \Bigg\{u_1(\boldsymbol{\pi}_1^{D,*},\bar{\boldsymbol{\pi}}_2^D) - u_1(\bar{\boldsymbol{\pi}}_1^D,\bar{\boldsymbol{\pi}}_2^D)\} - \delta_1 \sum_{a_1^{1:D}\in\mathcal{A}_1^P}\left[\prod_{d=1}^{D}\pi_1^{d,*}\left(a_1^d|a_1^{1:d-1}\right)\log\left(1/\boldsymbol{\tau}_1(a_1^{1:D})\right)\right]\Bigg\} \\
= &\max_{\boldsymbol{\pi}_1^{D,*}\in\Delta(A_1)} \{u_1(\boldsymbol{\pi}_1^{D,*},\bar{\boldsymbol{\pi}}_2^D) - u_1(\bar{\boldsymbol{\pi}}_1^D,\bar{\boldsymbol{\pi}}_2^D)\} - \lambda_1 \mathbb{E}_{a_1^{1:D}\sim\prod_{d=1}^{D}\pi_1^{D,*}\left(a_1^d|a_1^{1:d-1}\right)}\log\left(1/\boldsymbol{\tau}_1(a_1^{1:D})\right) \\
\geq &\max_{\boldsymbol{\pi}_1^{D,*}\in\Delta(A_1)} \{u_1(\boldsymbol{\pi}_1^{D,*},\bar{\boldsymbol{\pi}}_2^D) - u_1(\bar{\boldsymbol{\pi}}_1^D,\bar{\boldsymbol{\pi}}_2^D)\} - \lambda_1 \max_{a_1^{1:D}\sim\mathcal{A}_1^P}\log\left(1/\boldsymbol{\tau}_1\left(a_1^{1:D}\right)\right) \\
\geq &\max_{\boldsymbol{\pi}_1^{D,*}\in\Delta(A_1)} \{u_1(\boldsymbol{\pi}_1^{D,*},\bar{\boldsymbol{\pi}}_2^D) - u_1(\bar{\boldsymbol{\pi}}_1^D,\bar{\boldsymbol{\pi}}_2^D)\} - \lambda_1\delta_1
\end{aligned}
$$

The first inequality holds because the KL divergence is nonnegative by definition. The second inequality follows from the fact that the negative entropy function is nonpositive over the probability simplex. Finally, the third inequality is a direct consequence of the definition of $\delta_1$.

An analogous argument applies to Player 2, yielding the corresponding result.

$$
0 \geq \max_{\boldsymbol{\pi}_2^{D,*}\in\Delta(A_2)} \{u_2(\bar{\boldsymbol{\pi}}_1^D,\boldsymbol{\pi}_2^{D,*}) - u_2(\bar{\boldsymbol{\pi}}_1^D,\bar{\boldsymbol{\pi}}_2^D)\} - \lambda_2\delta_2
$$

Thus, the exploitability of $\bar{\boldsymbol{\pi}}_1^D$ is bounded by $\lambda_1\delta_1$, and similarly, the exploitability of $\bar{\boldsymbol{\pi}}_2^D$ is bounded by $\lambda_2\delta_2$. This directly establishes the desired result.

**Lower Bound of Q-value.** Assuming we have sequentially computed the Q-values for ground truth action $\left\{a_i^{1,*},\ldots,a_i^{d,*}\right\}$, we can derive a lower bound using the log-sum-exp inequality as follows:

$$
\begin{aligned}
Q_i^d(s,a_i^{1:d-1,*},a_i^{d,*}) &\triangleq \log\sum_{a_i^{d+1:D}}\exp\left\{\boldsymbol{Q}_i(s,a_i^{1:d,*},a_i^{d+1:D}) + \beta\log\boldsymbol{\tau}_i(a_i^{1:d,*},a_i^{d+1:D}|s)\right\} \\
&\geq \max_{a_i^{d+1:D}}\boldsymbol{Q}_i(s,a_i^{1:d,*},a_i^{d+1:D,*}) + \beta\log\boldsymbol{\tau}_i(a_i^{1:d,*},a_i^{d+1:D,*}|s) \\
&= \boldsymbol{Q}_i(s,a_i^{1:D,*}) + \beta\log\boldsymbol{\tau}_i(a_i^{1:D,*}|s),
\end{aligned}
$$

We first apply the log-sum-exp inequality to derive a tight lower bound for the decomposed Q-value $Q_i^d$. For the ground truth joint action $a_i^{1:D,*}$—which is obtained via search to maximize the objective $\boldsymbol{Q}_i + \beta\log\boldsymbol{\tau}_i$—this bound is achieved exactly. This implies that for any joint action $a_i^{1:D}$, the quantity $\boldsymbol{Q}_i(s,a_i^{1:D}) + \beta\log\boldsymbol{\tau}_i(a_i^{1:D}\mid s)$ serves as a valid and tight lower bound for $Q_i^d(s,a_i^{1:d})$. Leveraging this bound eliminates the need to compute the full log-sum-exp over the sub-action space, thereby significantly reducing computational complexity.

## C.1. Loss Function Derivation.

The training objective minimizes KL divergence between the defined learning objective and LLM's policy:

$$
\begin{aligned}
&= \arg\min_{\pi_\phi} \mathbb{E}_{(s,a_i^{1:d-1})\sim\mathcal{D}} \left[ D_{\mathrm{KL}} \left( \pi_i^{d,*}(\cdot|s,a_i^{1:d-1}) \| \pi_\phi(\cdot|s,a_i^{1:d-1}) \right) \right] \\
&= \arg\min_{\pi_\phi} \quad \mathbb{E}_{(s,a_i^{1:d-1})\sim\mathcal{D}} \left[ D_{\mathrm{KL}} \left( \frac{\exp\left\{ Q_i^d(s,a_i^{1:d-1},a_i^d) \right\}}{\sum_{a_i^d} \exp\left\{ Q_i^d\left(s,a_i^{1:d-1},a_i^d\right) \right\}} \middle\| \pi_\phi(\cdot|s,a_i^{1:d-1}) \right) \right] \\
&= \arg\min_{\pi_\phi} \quad \mathbb{E}_{(s,a_i^{1:d-1})\sim\mathcal{D}} \left[ \sum_{a_i^d \in \mathcal{A}_i^d} \frac{\exp\left\{ Q_i^d(s,a_i^{1:d-1},a_i^d) \right\}}{\sum_{a_i^d} \exp\left\{ Q_i^d\left(s,a_i^{1:d-1},a_i^d\right) \right\}} \log \frac{\pi_{\mathrm{unit}}^*\left(a_i^d|s,a_i^{1:d-1}\right)}{\pi_\phi\left(a_i^d|s,a_i^{1:d-1}\right)} \right] \\
&= \arg\max_{\pi_\phi} \quad \mathbb{E}_{(s,a_i^{1:d-1})\sim\mathcal{D}} \left[ \sum_{a_i^d \in \mathcal{A}_i^d} \frac{\exp\left\{ Q_i^d(s,a_i^{1:d-1},a_i^d) \right\}}{\sum_{a_i^d} \exp\left\{ Q_i^d\left(s,a_i^{1:d-1},a_i^d\right) \right\}} \log \pi_\phi\left(a_i^d|s,a_i^{1:d-1}\right) \right] \\
&= \arg\max_{\pi_\phi} \quad \mathbb{E}_{(s,a_i^{1:d-1})\sim\mathcal{D}} \left[ \sum_{a_i^d \in \mathcal{A}_i^d} \log \pi_\phi\left(a_i^d|s,a_i^{1:d-1}\right) \cdot \exp\left\{ Q_i^d(s,a_i^{1:d-1},a_i^d) \right\} \right] \\
&= \arg\max_{\pi_\phi} \quad \mathbb{E}_{(s,a_i^{1:d-1},a_i^d)\sim\mathcal{D}} \left[ \log \pi_\phi\left(a_i^d|s,a_i^{1:d-1}\right) \cdot \exp\left\{ Q_i^d(s,a_i^{1:d-1},a_i^d) \right\} \right]
\end{aligned}
$$

# D. Implementation details

## D.1. piKL-Hedge in Data Collection

During collecting data, We compute action-specific Q-values using the piKL-Hedge algorithm. The hyperparameters, primarily adopted from the original piKL-Hedge implementation (Jacob et al., 2022), are summarized in Table 3.

| Hyperparameter | Value |
|---|---|
| Search iterations | 256 |
| Number of candidate actions | 50 |
| Trade-off factor $\beta$ | 0.1 |
| Max candidate actions per unit | 6 |
| Nash explore ($\epsilon$) | 0.1 |
| Nash explore, S1901M | 0.1 |
| Nash explore, F1901M | 0.1 |

*Table 3.* Hyperparameters used for piKL-Hedge during data collection.

## D.2. Fine-tuning Details

We provide the hyperparameters used for training DipLLM in Table 4. Our model is built on the LLaMA 3 8B architecture as the backbone. During training, we employ the Low-Rank Adaptation (LoRA) method (Hu et al., 2022) to update the parameters of the entire LLM.

# E. Additional Experiments

## E.1. Performance of DipLLM Against Leading Large Language Models

We selected OpenAI-o3-mini and DeepSeek-R1 as opponents for further comparison with DipLLM. Our results demonstrate that even against models with strong reasoning capabilities, DipLLM exhibits remarkable superiority.

| Hyperparameter | Value |
|---|---|
| Optimizer | AdamW |
| LoRA $\alpha$ | 32 |
| Dropout Prob | 0.05 |
| Batch Size | 4 |
| Learning Rate Schedule | Linear |
| learning_rate | 2e-4 |
| Epoch | 5 |
| Adaptation | 16 |
| Max Seq. Len. | 2048 |

*Table 4.* The hyperparameters for fine-tuning DipLLM.

| Agent | SoS Scores↑ | Win Rate↑ | Most SC↑ | Survived↑ | Defeated↓ |
|---|---|---|---|---|---|
| **DipLLM (Ours)** | **60.7%**±0.5% | **69.7%**±0.7% | **68.2%**±0.6% | **27.1%**±0.1% | **3.2%**±0.1% |
| OpenAI-o3-mini | 16.3%±0.3% | 13.9%±0.2% | 11.7%±0.2% | 51.9%±0.5% | 34.2%±0.2% |
| Deepseek-R1 | 14.6%±0.1% | 10.8%±0.1% | 8.5%±0.1% | 49.3%±0.3% | 39.9%±0.7% |
| GPT-4o | 3.7%±0.1% | 1.8%±0.2% | 2.5%±0.1% | 40.2%±0.3% | 57.9%±0.6% |

*Table 5.* Performance of different agents in a population including DipLLM and top large language models.

## E.2. Enhancing DipLLM with Online Search

To further evaluate DipLLM's potential, we conducted additional experiments during the rebuttal period by integrating a lightweight search mechanism in 1v6 settings against DipNet. The results show a clear performance gain, indicating that DipLLM can benefit from search when resources permit. These findings highlight the model's strong decision-making ability and suggest that incorporating online reasoning could further enhance its capabilities.

| DipLLM rollouts | SoS Scores↑ | Win Rate↑ | Most SC↑ | Survived↑ | Defeated↓ |
|---|---|---|---|---|---|
| $n = 0$ | 16.7%±0.1% | 13.1%±0.2% | 17.5%±0.2% | 47.1%±0.1% | 38.3%±0.2% |
| $n = 5$ | 18.1%±0.2% | 14.7%±0.2% | 19.3%±0.2% | **47.8%**±0.4% | 37.5%±0.2% |
| $n = 10$ | **20.2%**±0.1% | **15.6%**±0.2% | **21.6%**±0.2% | 47.4%±0.3% | **37.0%**±0.1% |

*Table 6.* Results of DipLLM with different numbers of equilibrium search rollouts, playing against Cicero.

## E.3. Preference Learning in Diplomacy

We explored the use of preference learning by labeling data based on outcome-based reward magnitudes and applying Direct Preference Optimization (DPO) (Rafailov et al., 2023). However, this approach yielded unsatisfactory results. A key difficulty lies in the ambiguous nature of preferences in multi-agent strategic settings—especially when dealing with joint actions or long-term planning. This ambiguity leads to noisy annotations and poor learning signals, ultimately limiting optimization performance. We also adopt a reward-based learning framework and compare it to standard policy optimization methods such as Proximal Policy Optimization (PPO) (Schulman et al., 2017). Experimental results show that our method outperforms PPO, demonstrating the effectiveness of leveraging structured game rewards over noisy preference annotations.

| Method | SoS Scores↑ | Win Rate↑ | Most SC↑ | Survived↑ | Defeated↓ |
|---|---|---|---|---|---|
| **Ours** | **29.3%**±0.2% | **25.2%**±0.2% | **29.3%**±0.2% | **45.8%**±0.3% | **29.0%**±0.1% |
| PPO | 23.4%±0.2% | 20.2%±0.2% | 23.4%±0.2% | 45.4%±0.5% | 34.4%±0.2% |
| DPO | 1.2%±0.1% | 0.7%±0.1% | 1.3%±0.1% | 38.2%±0.3% | 61.1%±0.7% |

*Table 7.* Performance of LLM agents fine-tuned using different methods against six DipNet agents.

