# OpenReview forum: "DipLLM: Fine-Tuning LLM for Strategic Decision-making in Diplomacy"
_ICML.cc/2025/Conference — ICML 2025 poster_

### Official Review · Reviewer_vcKR · 2025-03-13

**Overall Recommendation:** 3

**Summary:**

This work proposes to fine-tune LLM with a small amount of data to achieve strong performance in Diplomacy. They propose to factorize the combinatorial joint action space into manageable subspaces in an autoregressive manner, and derive a corresponding learning objective for the factorized actions to fine-tune the LLM. The fine-tuned LLM outperforms various agents including Cicero using a small amount of fine-tune data.

## update after rebuttal
The authors' response has addressed my concern. I'll keep my score.

**Claims And Evidence:**

Most of the claims are clear and convincing. My main concern is explained in detail in the theoretical claim part.

**Essential References Not Discussed:**

No.

**Experimental Designs Or Analyses:**

The experiment is extensive and valid. The authors compare their method with various methods including SOTA agents like Cicero. They also provide detailed ablations and case studies to analyze their method.

**Methods And Evaluation Criteria:**

This submission is well-motivated by the combinatorial action space of Diplomacy and the inherent limitation of prompt-based LLM agents. The proposed autoregressive factorization method for fine-tuning is intuitive and effective.

**Other Comments Or Suggestions:**

N.A.

**Other Strengths And Weaknesses:**

N.A.

**Questions For Authors:**

This work is well-motivated. The proposed method is intuitive and effective. And the empirical experiment results are strong and valid.

My main concern is about the theoretic justification for the factorization in Eq. (3), which is discussed in detail in the "Theoretical Claims" part of my review. The soundness of this equation is of great importance because it is the foundation of subsequent theoretic results. The answer to this question would influence my final assessment of this work.

**Relation To Broader Scientific Literature:**

This paper proposes to fine-tune LLM in specific games for strategic decision-making. This approach can be generally applied to many other domains. It also provides a new competitive agent for Diplomacy, which is a common game used for studying language and strategic play.

**Theoretical Claims:**

My main concern is about the theoretical justification for the factorization of the Q-function in Eq. (3).

> Claim 1 (L208-209): $\mathbf{Q}_i$ represents the **reward** for the joint action.

The factorization of Eq. (2) is built on piKL-Hedge from Eq. (1). In the piKL-Hedge paper [1] and in standard RL literature [2], $Q_i(s, a)$ is the expected future reward for player $i$ from playing action $a$ in state $s$, not the reward for the joint action, which is $r_i(s, a)$. The same problems exist in Eq. (1) (L150) where the $Q$ is used without definition, and in Eq. (3) (L165) where $Q_i^d$ is defined to "represent the **reward** of action $a_i^d$".

Moreover, for multi-player games, the Q function not only depends on the state and action of the current player, but also depends on the actions or policies of other players [3]. The non-standard and unclear definition of $Q$ makes it hard to verify the correctness of factorization in Eq. (3).

[1] Jacob, Athul Paul, et al. "Modeling strong and human-like gameplay with KL-regularized search." International Conference on Machine Learning. PMLR, 2022.

[2] Sutton, Richard S., and Andrew G. Barto. Reinforcement learning: An introduction. Vol. 1. No. 1. Cambridge: MIT press, 1998.

[3] Littman, Michael L. "Markov games as a framework for multi-agent reinforcement learning." Machine learning proceedings 1994. Morgan Kaufmann, 1994. 157-163.

> Claim 2 (L215): $\frac{1}{D}\sum_{d=1}^D Q_i^d(s, c_i^{1:d-1},a_i^d) = \mathbf{Q}_i(s, a_i^{1:D})$

This factorization is the key equation used in subsequent theoretic results but it is not justified. If the Q here is the expected future reward as defined in piKL-Hedge, it is not trivial to see why this factorization holds. Some works in MARL like [4, 5] also discuss decomposition in autoregressive or sequential policy, and their decompositions are usually in the form of expectation over policies. I would suggest the authors theoretically justify why this factorization holds since it is the core equation in their method.

[4] Kuba, Jakub Grudzien, et al. "Trust region policy optimisation in multi-agent reinforcement learning." arXiv preprint arXiv:2109.11251 (2021).

[5] Fu, Wei, et al. "Revisiting some common practices in cooperative multi-agent reinforcement learning." arXiv preprint arXiv:2206.07505 (2022).

> Claim 3 (L217): $\tau_i^d(a_i^d|s,c_i^{1:d-1}) = \mathbf{\tau}(a_i^{1:D}|s)$

I think there might be a missing $\Pi_{d=1}^D$ on the left-hand side of this equation. The probability of the joint action should be the product of the probability of each sub-action. I would suggest the authors check other potential typos in their manuscript.

---

> ### Author Rebuttal · Authors · 2025-03-31
>
> **Q1. Explain the definition of the original joint Q-value.**
>
> A1. In the context of Diplomacy, we define the joint Q-value as the **expected cumulative reward** a player receives after executing a given joint action from the current state [1]. When applying **iterative equilibrium search methods** such as piKL-Hedge, this expected cumulative reward can be computed using the following equation:
>
> $$\boldsymbol{Q}\_i(s,a_i)=\mathbb{E}\_{\boldsymbol{a}\_{-i}\sim\boldsymbol{\pi}\_{-i}(\cdot|s)}[u_i(s,a_i,\boldsymbol{a}\_{-i})]$$
> where $\pi_{-i}$ ​represents the strategy followed by other players for their joint actions, and $u_{i}$​ denotes the utility function after completing the search under the current state and all players' actions.
>
> ---
> **Q2. Explain the theoretical justification for factorization.**
>
> A2. We appreciate your attention to the theoretical justification of our factorization approach. Unlike HAPPO, which first defines decomposed A-values and then establishes equations they satisfy, **our factorization equation serves as a flexible credit assignment condition rather than a strict definition**. It ensures that as long as the total value of $D \cdot Q$ is distributed across different unit actions, the factorization remains valid. Any method that adheres to this assignment rule does not affect our theoretical guarantees or experimental results.
>
> While multiple approaches can satisfy this condition, due to character limits, we provide **a simple example inspired by Q-Transformer** [2].
>
> $$Q_i\left(s,c_i^{1:d-1},a_i^d\right)\triangleq\begin{cases}
> \mathbb{E}\_{a_i^{d+1}\sim\pi\_i^{d+1}}\left[Q\_i\left(s,c_i^{1:d},a\_i^{d+1}\right)\right],&\text{if }d\in\\{1,\ldots,D-2\\}\\\\
> \mathbb{E}\_{a_i^{d+1}\sim\pi_i^{d+1}}\left[\boldsymbol{Q}\_i\left(s,[a_i^{1:d},a\_i^{d+1}]\right)\right],&\text{if }d=D-1\\\\
> D\cdot\boldsymbol{Q}\_i\left(s,a\_i^{1:D}\right)-\sum_{d=1}^{D-1}Q\_i^d\left(s,c\_i^{1:d-1},a\_i^d\right),&\text{if }d=D
> \end{cases}$$
>
> The intuition behind this formulation is that except for the last two actions (d=D, D-1), the **Q-value is the expected value under the decomposed policy for the next dimension’s actio**n. The penultimate dimension (d=D-1) is assigned the **expected joint Q-value over the remaining action**. Finally, the terminal dimension’s value is determined by the joint Q-value of the complete action sequence. Based on this definition, we now provide a formal proof to justify this factorization.
>
> *Proof*. For any joint action $a_i^{1:D}=\\{a_i^1,a_i^2,\ldots,a_i^D\\}$ and its decomposed action components, the following identity holds:
>
> $$\begin{aligned}
> &\sum\_{d=1}^DQ^d_i\left(s,c_i^{1:d-1},a_i^d\right)\\\\
> &=\sum\_{d=1}^{D-1}Q^d_i\left(s,c\_i^{1:d-1},a\_i^d\right)+Q^D_i\left(s,c\_i^{1:D-1},a\_i^D\right)\\\\
> &=\sum\_{d=1}^{D-1}Q^d_i\left(s,c\_i^{1:d-1},a\_i^d\right)+\left[D\cdot\boldsymbol{Q}\_i\left(s,a\_i^{1:D}\right)-\sum\_{d=1}^{D-1}Q_i^d\left(s,c\_i^{1:d-1},a_i^d\right)\right]\\\\
> &=D\cdot\boldsymbol{Q}\_i\left(s,a\_i^{1:D}\right)
> \end{aligned}$$
>
> We also provide a **rigorous fundamental definition of decomposed Q-values** and formally prove their soundness and correctness **using Bayes' theorem and the log-sum-exp inequality**. Due to space constraints, we are unable to present the full details here, but we would be happy to include them in the second round of the rebuttal if you are interested.
>
> ---
> **Q3. Explain the equation for $\tau_i$ in Equation 3 (L217).**
>
> A3. Our formulation distributes the joint anchor policy $\boldsymbol{\tau}_i$​ across the factored unit actions. The key property of this distribution is that each unit’s $\tau_i$​ probability remains aligned with the joint $\boldsymbol{\tau}_i$, ensuring consistency. This relationship can be formally expressed as:
> $$\tau_i\left(a_i^d|s,c_i^{1:d-1}\right)\triangleq\begin{cases}
> \tau_i\left(a_i^{d+1}|s,c_i^{1:d}\right),&\text{if }d\in\\{1,\ldots,D-1\\}\\\\
> \boldsymbol{\tau}_i\left(a_i^{1:D}|s\right),&\text{if }d=D
> \end{cases}$$
>
> This formulation leads to the result that $\prod_{d=1}^D\tau_i\left(a_i^d|s,c_i^{1:d-1}\right)\neq\boldsymbol{\tau}_i(a_i^{1:D}|s)$. However, this discrepancy has no impact on our approach because our primary concern is not whether the factored $\tau_i$ precisely reconstructs $\boldsymbol{\tau}_i$, but rather whether the original and factored policies $\boldsymbol{\pi}_i$ remain equivalent.
>
> ---
> `Summary Response`
>
> We sincerely appreciate your attention to the theoretical justification for factorization. Your insights help strengthen our theoretical analysis and provide valuable directions for future research. Please let us know if our response has adequately addressed your concerns. We would be delighted to engage in further discussions to refine and improve our manuscript.
>
> [1] Mastering the game of no-press diplomacy via human-regularized reinforcement learning and planning. ICLR, 2023.
>
> [2] Q-transformer: Scalable offline reinforcement learning via autoregressive q-functions. CoRL, 2023.

---

### Official Review · Reviewer_FG1V · 2025-03-13

**Overall Recommendation:** 3

**Summary:**

This paper introduces DipLLM, a fine-tuned Large Language Model (LLM) designed for strategic decision-making in the game of Diplomacy. The authors argue that traditional equilibrium search methods require substantial computational resources, whereas fine-tuning an LLM can yield superior performance with significantly less data. The proposed DipLLM employs an autoregressive factorization framework to break down complex multi-unit action assignments into sequential unit-level decisions. The learning objective is designed to approximate the Nash equilibrium, and fine-tuning is performed using a Diplomacy-specific dataset. Empirical results demonstrate that DipLLM outperforms Cicero—a state-of-the-art Diplomacy agent—while requiring only 1.5% of Cicero’s training data. The authors also provide theoretical analysis to establish the equivalence and optimality of their approach.

**Claims And Evidence:**

The author claims to use only 1.5% of the data required by the state-of-the-art Cicero model. However, comparing the size of data between an online reinforcement learning (RL) method and an offline supervised learning method is inherently flawed and lacks meaningful insight. Online RL methods typically require more data due to their exploration and iterative learning nature, whereas supervised methods rely on pre-collected, labeled datasets.

**Essential References Not Discussed:**

No.

**Experimental Designs Or Analyses:**

Yes

**Methods And Evaluation Criteria:**

Yes

**Other Comments Or Suggestions:**

See above.

**Other Strengths And Weaknesses:**

#### Strengths
- Strong empirical performance: Outperforms Cicero while using only 1.5% of its training data.
- Theoretical grounding: Provides formal proofs for the equilibrium properties of the method.
- Well-structured experimental evaluation: Includes ablation studies and comparisons with baseline methods.
#### Weaknesses
- Data reliance: The approach still depends on externally generated datasets.
- Limited generalization: The method is only evaluated on no-press Diplomacy.

**Questions For Authors:**

No questions.

**Relation To Broader Scientific Literature:**

The paper is well-situated in the context of:

- Multi-agent reinforcement learning (e.g., equilibrium search methods like Cicero).
- LLM-based decision-making.
- Strategic AI in complex games (e.g., Go, Poker, Diplomacy).

**Theoretical Claims:**

Yes

---

> ### Author Rebuttal · Authors · 2025-03-31
>
> Detailed tables are available at https://sites.google.com/view/dipllm.
>
> **Q1. The approach still depends on externally generated datasets.**
>
> A1. Our approach relies on externally generated data to enable efficient data collection and significantly accelerate LLM training. While self-play is a viable alternative that does not require external data, **training LLMs from scratch through self-play is impractical, as it requires the base model to already possess strong reasoning abilities and substantial computational resources to generate data [1]**. However, unfine-tuned LLMs perform poorly in Diplomacy, leading to low-quality data when interacting with the environment. To overcome this, we leverage DipNet to generate higher-quality training data, making the process more efficient.
>
> To evaluate whether DipLLM can further refine its strategy through self-play, we conducted an experiment in which the fine-tuned DipLLM generated data via self-play and underwent a second round of fine-tuning. As shown in Table R3, this led to further performance improvements, highlighting the potential of self-play. **With advances in LLM inference efficiency and capabilities, iterative self-play could become a viable alternative, reducing reliance on externally generated datasets.**
>
> Agent|Score
> -|-
> DipLLM|29.3±0.2
> **DipLLM+self-play**|**31.6±0.1**
>
> ---
> **Q2. Why is the method only evaluated on no-press Diplomacy.**
>
> A2. Diplomacy is widely recognized as a complex multi-agent benchmark, following Chess, Go, and Poker. It presents more difficult  challenges due to its **immense combinatorial action space**—with up to $10^{64}$ possible choices per turn—stemming from its mechanics, where each player controls up to 34 units, each with 26 possible actions.
>
> This complexity, compounded by **intricate player interactions**, makes Diplomacy a particularly demanding testbed for AI decision-making. Just as prior work on Poker (e.g., **Libratus [2]**) and Go (e.g., **AlphaGo [3]**) focused exclusively on their respective domains, existing Diplomacy research—including **DipNet [4], Brbot [5], SearchBot [6] and DORA [7] —has only evaluated on no-press Diplomacy.** Our evaluation follows this standard to make sure that comparisons with well-known standards are fair and useful.
>
>
> ---
> **Q3. Why compare data size between online RL methods and DipLLM (offline)?**
>
> A3. Our primary motivation for comparing data size is to highlight the **cost efficiency** of our method. Cicero, an off-policy RL method, follows a structured data collection pipeline:
>
> 1. Generating candidate action sets using its RL policy,
>
> 2. Refining actions through equilibrium search,
>
> 3. Interacting with the environment and storing the generated trajectories in a **large replay buffer**
>
> Our method follows a similar process, utilizing DipNet and equilibrium search to generate and refine candidate actions, which are then stored as an **offline dataset**. Given these similarities, comparing data size remains meaningful.
>
> As you pointed out, online RL has low sample efficiency and requires substantial computational resources, which motivates our choice of offline training for LLMs. Regarding computational cost, **Cicero requires 444 GPUs** running for an entire week to generate data. In contrast, **our approach achieves superior performance with just 8 GPUs**, demonstrating a clear advantage in both efficiency and cost.
>
> Moreover, Cicero and DNVI are not purely online RL methods but instead combine supervised learning (SL) with off-policy RL. **Both methods initially pretrain on tens of thousands of human expert games using SL before RL training.** This further highlights the efficiency of our method, as we achieve superior results with significantly lower data and computational costs.
>
> ---
> `Summary Response`
>
> Thank you for your valuable comments, which have helped us refine our experimental evaluation and gain deeper insights for future research. We are honored to have **your recognition of our paper’s structure, theoretical analysis, and empirical performance**. Please let us know if we have adequately addressed your concerns—we would be happy to engage in further discussions to improve our manuscript.
>
> [1] Learning to Reason with LLMs. OpenAI, 2024
>
> [2] Superhuman AI for heads-up no-limit poker: Libratus beats top professionals. Science, 2018.
>
> [3] Mastering the game of Go with deep neural networks and tree search. Nature, 2016.
>
> [4] No-press diplomacy: Modeling multi-agent gameplay. NeurIPS, 2019.
>
> [5] Learning to Play No-Press Diplomacy with Best Response Policy Iteration. NeurIPS, 2020.
>
> [6] Human-Level Performance in No-Press Diplomacy via Equilibrium Search. ICLR, 2021.
>
> [7] No-Press Diplomacy from Scratch. NeurIPS, 2021.

---

### Official Review · Reviewer_ELUa · 2025-03-14

**Overall Recommendation:** 3

**Summary:**

DipLLM is a fine-tuned LLM designed to play the complex multiplayer game Diplomacy. DipLLM leverages an autoregressive factorization framework to simplify multi-unit action assignments into unit-level decisions. By fine-tuning with only 1.5% of the data needed by the state-of-the-art Cicero model, DipLLM achieves superior performance, demonstrating the potential of LLMs in complex strategic decision-making in multiplayer games.

**Claims And Evidence:**

1、Equations (1) and (2) are in identical forms. The primary innovation by the authors is the Autoregressive Factorization, which decomposes a reward objective into multiple action objectives. The paper's contribution appears to be incremental. The authors should clarify their specific contributions to distinguish their work from existing methods in the learning objective
2、In Figure 3, raw data is collected by interacting with the environment via DipNet. Q-values are generated through piKLHedge search, and the data is stored in prompt form.  DipLLM seems to enhance the model by generating offline data and using rewards. Why not use preference optimization, which might be more suitable than SFT for this purpose? The authors should consider discussing the rationale behind their chosen approach.

**Essential References Not Discussed:**

I have no comments.

**Experimental Designs Or Analyses:**

1.	Table 1 does not compare with the latest method, Richelieu; the comparison with Richelieu only appears in Figure 4.
2.	Figure 4 is confusing; the reasons for the scores on the horizontal and vertical axes are unclear.
3.	The improvement in metrics is not compelling. For example, DipLLM achieved 50.3%±0.7% in Survived, while Cicero achieved 50.1%±0.5%.
4.	Compared to the best large language models (e.g., reasoning models like OpenAI-o3, DeepSeek-R1), does DipLLM have better performance?

**Methods And Evaluation Criteria:**

1、The single action does not have a specified reward; Qi represents the reward for the joint action. Is this setup reasonable? Could SFT lead to overfitting on the joint action's final objective?
2、In the Data Collection part, for any joint action, the unit-level values are set equal to the original joint value. This setup seems unreasonable; should different weights be assigned?
3、What are the details of the data collection and evaluation processes? Is there a risk of data leakage?

**Other Comments Or Suggestions:**

I have no comments.

**Other Strengths And Weaknesses:**

I have no comments.

**Questions For Authors:**

1、DipLLM seems to enhance the model by generating offline data and using rewards. Why not use preference optimization, which might be more suitable than SFT for this purpose? The authors should consider discussing the rationale behind their chosen approach.
2、In the Data Collection part, for any joint action, the unit-level values are set equal to the original joint value. This setup seems unreasonable; should different weights be assigned?
3、What are the details of the data collection and evaluation processes? Is there a risk of data leakage?
4、Table 1 does not compare with the latest method, Richelieu; the comparison with Richelieu only appears in Figure 4.
5、Compared to the best large language models (e.g., reasoning models like OpenAI-o3, DeepSeek-R1), does DipLLM have better performance?

**Relation To Broader Scientific Literature:**

I have no comments.

**Theoretical Claims:**

I have no comments.

---

> ### Author Rebuttal · Authors · 2025-03-31
>
> Detailed tables are available at https://sites.google.com/view/dipllm.
>
> **Q1. Why define rewards only for joint actions but not for single actions?**
>
> A1. **This setup follows prior work** on Diplomacy [1], where a player's action is the decisions of all units, with rewards based on the resulting state. As it is difficult to **quantify the contribution of each unit action**, we maintain the setup of defining rewards at the joint action, allowing the model to learn proper credit assignment on its own.
>
> ---
> **Q2. Why not use preference learning to optimize the LLM model?**
>
> A2. Preference learning requires high-quality data, but **manual annotation is costly** [2]. Diplomacy’s complexity further complicates data collection. Our attempt to **label preferences by reward magnitude and train with DPO (Table R4) failed**, likely due to unclear preference definitions for factored actions.
>
> Conversely, tasks with well-defined reward functions, like Go and Diplomacy, are better **suited for reward optimization**. Diplomacy’s reward function, defined by game scores, enables effective optimization. To highlight our method's advantages, we compared it to the **reward-based algorithm PPO**. Table R4 shows our approach outperforms.
>
> Method|Score
> -|-
> **Ours**|**29.3±0.2**
> PPO|23.4±0.2
> DPO|1.2±0.1
> ---
> **Q3: Why not assign different weights to different unit actions?**
>
> A3. We considered weighting but prioritized aligning the joint policy with the Nash equilibrium over individual unit action distributions. Theorem 1 (L180-L183) shows that if Equation 3 holds, the joint decomposed policy matches the original approximate Nash equilibrium strategy. Thus, **weighting may alter unit action distributions but not the overall joint strategy.**
>
> ---
> **Q4. Explain the details of the data collection and evaluation processes. Any risk of data leakage?**
>
> A4. Due to space limits, we refer you to our response to Reviewer mCms, Q4, for data collection details. To evaluate model performance, we use **Meta's offline Diplomacy environment** and an opponent pool. In each game, two agents are randomly sampled in a 1v6 setup—one controls a single power, while identical copies of the opposing agent control the other six powers. There is **no risk of data leakage**, as evaluation is based on adversarial gameplay rather than a fixed test set.
>
> ---
> **Q5. Explain the meanings for the scores in Figure 4.**
>
> A5. Figure 4's scores show each **agent's (y-axis) sum-of-squares score** when competing against six copies of another agent (x-axis).
>
> ---
> **Q6. Why is Richelieu only compared in Figure 4, not Table 1?**
>
> A6. Since Richelieu's code was unavailable despite an open-sourced repository, Figure 4 (L299) shows the **Prompt (Richelieu) agent performed significantly worse**. Including it in Table 1 would **inflate other agents' scores**, so we excluded it for fairness. To address your concerns, Table R5 includes results with Richelieu's opponent pool.
>
> ---
> **Q7. Does DipLLM outperform top LLMs (e.g., OpenAI-o3, DeepSeek-R1)?**
>
> A7. With O3 unavailable, we tested DipLLM against OpenAI-o3-mini and DeepSeek-R1. Table R6 shows **DipLLM's superiority over strong reasoning models**, which struggle with the vast action space.
>
> Agent|Score
> -|-
> **DipLLM**|**60.7±0.5**
> OpenAI-o3-mini|16.3±0.1
> Deepseek-R1|14.6±0.1
> GPT4o|3.7±0.1
>
> ---
> **Q8. Clarify contributions to distinguish DipLLM from previous work.**
>
> A8. We are the **first to fine-tune LLMs for Diplomacy**. Previous methods either trained small, specific models on large-scale data or used prompt-driven LLMs. In contrast, we fine-tune LLMs with relatively **small data to approximate Nash equilibrium strategies**, achieving superior performance. To achieve this, we introduced an autoregressive factorization framework and a theoretically grounded fine-tuning approach within this framework.
>
> ---
> **Q9. Could SFT lead to overfitting on the joint action's final objective?**
>
> A9. No, our fine-tuning does not cause overfitting to the joint action's final objective. Unlike standard SFT, our objective is to **minimize KL divergence to an approximate Nash equilibrium**, ensuring alignment with the equilibrium distribution rather than mere action imitation. Our ablation study (L416-430) confirms this, showing in Figure 7 that the model **learns the target distribution** instead of overfitting those actions.
>
> ---
> **Q10. The improvement in metrics is not compelling, especially in Survived.**
>
> A10. Survived is analogous to a **draw** in chess or Go and not strictly a **"higher is better"** metric. When win rate rises while defeat rate stays constant, survival naturally decreases. **DipLLM, using only 1.5% of Cicero’s training data, achieved a higher win rate and lower defeat rate with a similar survival rate**, demonstrating stronger strategic performance.
>
> [1] Mastering the Game of No-Press Diplomacy via Human-Regularized Reinforcement Learning and Planning. ICLR 2023
>
> [2] A Survey of Direct Preference Optimization. ArXiv 2025

---

### Official Review · Reviewer_mCms · 2025-03-19

**Overall Recommendation:** 3

**Summary:**

This paper introduces DipLLM, a fine-tuned Large Language Model (LLM) aimed at learning equilibrium policies for the no-press variant of the game of Diplomacy. The authors propose an autoregressive factorization framework to break down multi-unit action selection into smaller, sequential decisions, thereby mitigating the combinatorial explosion in action space. They then define a learning objective akin to the final policy in piKL-Hedge, showing its theoretical equivalence and approximate optimality for two-player zero-sum settings. Leveraging a small subset of Diplomacy gameplay data (1.5% of Cicero’s dataset), DipLLM reportedly achieves superior performance to the state-of-the-art Cicero model, all while requiring significantly fewer computational and data resources.

**Claims And Evidence:**

DipLLM can approximate a Nash-like equilibrium policy via autoregressive factorization and fine-tuning.

**Essential References Not Discussed:**

N.A.

**Experimental Designs Or Analyses:**

he experiments compare DipLLM to well-known baselines, including Cicero.

**Methods And Evaluation Criteria:**

Their main evaluation strategy uses 1v6 tournaments against strong baselines, which has been used in Diplomacy research to assess individual agent strength in a multi-agent environment.

**Other Comments Or Suggestions:**

M1ore details on the transition from DipNet data to the final autoregressive factorization format would be helpful.

**Other Strengths And Weaknesses:**

Strengths:
The explanation of autoregressive factorization is well-presented. Including both theoretical and empirical justification is commendable.

Weaknesses:
While the data usage is small, it is still anchored in offline or external equilibrium-search-based generation (from DipNet + piKL-Hedge). A purely self-play iteration might be tested to illustrate robust self-improvement or autonomy.

Although the authors show that Cicero’s performance improves with search rollouts, they do not attempt to incorporate an online search procedure for DipLLM, which might reinforce or surpass results further.

**Questions For Authors:**

1.  How critical is the general knowledge from the base LLM (e.g., on language or reasoning tasks) to the success in Diplomacy? Would a specialized but smaller model, with the same data, still fall short?

2.  Could DipLLM be improved further through repeated self-play to generate data and refine its policy iteratively (rather than relying on another anchor model)?

**Relation To Broader Scientific Literature:**

Overall, the paper appears consistent with, and adds to, an emerging literature combining large foundation models with multi-agent game theory.

**Theoretical Claims:**

The joint policy formed by multiplying the factorized distributions can match the original distribution from piKL-Hedge.

---

> ### Author Rebuttal · Authors · 2025-03-31
>
> Detailed tables are available at https://sites.google.com/view/dipllm.
>
> **Q1. How crucial is base LLM knowledge for Diplomacy? Would a smaller, specialized model with the same data still underperform?**
>
> A1. Our experiments show that **general knowledge in base LLMs is crucial** for success in Diplomacy, and specialized smaller models consistently underperform.
>
> First, comparative experiments with Llama models (Appendix E.1, Table 5) show stronger bases (Llama-3-8B) outperform weaker ones (Llama-2-7B) when fine-tuned, highlighting the value of broader knowledge.
>
> Llama|Score
> -|-
> **8B-distill**|**31.8±0.2**
> 8B|29.3±0.1
> 7B|27.9±0.1
>
> Second, to further validate this, we conducted additional experiments with Qwen models of various sizes, including standard and knowledge-distilled variants. Table R1 shows that (1) **larger models outperform smaller ones** and (2) among similar-sized models, **knowledge-distilled versions (e.g., DeepSeek) perform better**. This confirms that both model scale and knowledge quality significantly impact performance. The table below shows partial results.
>
> Qwen2.5|Score
> -|-
> **1.5B-distill**|**27.7±0.1**
> 3B|27.0±0.2
> 1.5B|23.1±0.1
> 0.5B|20.3±0.1
>
> Finally, our comparison with DipNet (§ 5.2, Figure 6) shows that as data increases, both improve, but LLMs leverage general knowledge for superior data efficiency. At 500 games, DipLLM outperforms DipNet by 6.7%, while **fine-tuning smaller specific models yields only marginal gains**.
>
> ---
>
> **Q2. Why didn’t DipLLM use an online search like Cicero, and how does this impact performance?**
>
> A2. As noted in the **Limitations** section, equilibrium search demands **substantial computational resources** (192–448 GPUs) and incurs high inference costs, making its integration with LLMs highly expensive. Despite this, DipLLM outperforms Cicero, even when Cicero utilizes a limited number of search iterations (§5.2, Figure 5).
>
> To further explore DipLLM’s potential, we integrated equilibrium search during the rebuttal period. Given computational constraints, we conducted as many preliminary experiments as possible. Our evaluation (Table R2) shows that **incorporating search improves decision quality (+3.5% win rate, rollouts=10)**, demonstrating DipLLM’s capacity to leverage enhanced reasoning while preserving sample efficiency.
>
> Rollouts|Score
> -|-
> 0|16.7±0.1
> 5|18.1±0.2
> **10**|**20.2±0.1**
> ---
> **Q3. Why doesn't DipLLM use pure self-play? Could iterative self-play improve it?**
>
> A2. **Pure self-play requires the base model to already possess strong reasoning abilities and substantial computational resources to generate data [1]**. Due to the complexity of Diplomacy, unfine-tuned LLMs perform poorly, resulting in low-quality data. To overcome this, we leverage DipNet to generate higher-quality training data, making the process more efficient.
>
> The suggestion to **improve DipLLM through repeated self-play is valuable**. To evaluate this, we use fine-tuned DipLLM incorporating equilibrium search to generate 100 games for further fine-tuning. As shown in Table R3, performance **improved by 2.3% after just one iteration**, indicating that multiple rounds could be beneficial. However, computational constraints limit scalability. Future work may explore parallelized self-play or distillation to enhance efficiency.
>
> Agent|Score
> -|-
> DipLLM|29.3±0.2
> **DipLLM+self-play**|**31.6±0.1**
> ---
> **Q4. Explain details on the transition from DipNet data to the final autoregressive factorization format.**
>
> A4. Our data collection process consists of the following steps:
> 1. **Generating Raw Data**:
> DipNet interacts with the Diplomacy environment as player $i$, producing raw data: game states $s$, joint actions $a_i^{1:D}$, and action probability values $\boldsymbol{\tau}_i(a_i^{1:D}|s)$.
> 1. **Computing Joint Q-Values**: The raw data is processed using the piKL-Hedge algorithm to compute Q-values $\boldsymbol{Q}(s,a_i^{1:D})$ for joint actions.
> 2. **Action Decomposition**: Joint actions are decomposed into unit-level actions, with Q-values $Q_i^d(·,a_i^d)$ and $\tau_i^d(a_i^d|·)$ assigned to each unit action via Equation 3.
> 3. **Textual Formatting**: Game states and decomposed actions are converted to text. Each unit action is stored as ground truth $a_i^d$ sequentially with preceding actions recorded as context $c_i^{1:d-1}$.
> 4. **Final Data Storage**: Data is stored in the transition  $(s,c_i^{1:d-1},a_i^d,Q_i^d,\tau_i^d)$.
>
> For more details on the stored data, please refer to Appendix B: Full Prompts for DipLLM.
>
> `Summary Response`
>
> Thanks for your valuable comments, which helped us strengthen our experimental evaluation. Our results show that **a stronger backbone (Table R1), online search (Table R2), and self-play (Table R3) all further enhance our model's performance**. Please let us know if we have sufficiently addressed your concerns—we would be happy to engage in further discussions to improve our manuscript.
>
> [1] Learning to Reason with LLMs. OpenAI, 2024

---

### Decision · Program_Chairs · 2025-05-01

**Decision:**

Accept (poster)

**Comment:**

The authors propose fine-tuning an LLM to play no-press Diplomacy and demonstrate superior performance to prior SOTA. Their technique uses a piKL-Hedge inspired learning objective that they show how to efficiently factorize over Diplomacy’s large combinatorial action space. The reviewers all found the approach clearly explained. While DipLLM outperforms prior approaches, it does so by building on top of pre-trained models (LLMs, DipNet) that have previously consumed large amounts of data making the strict training-data comparison a bit unfair. Lastly, I will note that the authors made a significant effort to address all reviewers’ concerns in the rebuttal process.